# OligoGym: Curated Datasets and Benchmarks for Oligonucleotide Drug Discovery

**Rachapun Rotrattanadumrong**[*]
Pharma Research and Early Development
F. Hoffmann-La Roche
rachapun.rotrattanadumrong@roche.com

**Carlo De Donno**[*]
Pharma Research and Early Development
F. Hoffmann-La Roche
carlo.de_donno@roche.com

## Abstract

Oligonucleotide therapeutics offer great potential to address previously undruggable targets and enable personalized medicine. However, their progress is often hindered by insufficient safety and efficacy profiles. Predictive modeling and machine learning could significantly accelerate oligonucleotide drug discovery by identifying suboptimal compounds early on, but their application in this area lags behind other modalities. A key obstacle to the adoption of machine learning in the field is the scarcity of readily accessible and standardized datasets for model development, as data are often scattered across diverse experiments with inconsistent molecular representations. To overcome this challenge, we introduce OligoGym, a curated collection of standardized, machine learning-ready datasets encompassing various oligonucleotide therapeutic modalities and endpoints. We used OligoGym to benchmark diverse classical and deep learning methods, establishing performance baselines for each dataset across different featurization techniques, model configurations, and splitting strategies. Our work represents a crucial first step in creating a more unified framework for oligonucleotide therapeutic dataset generation and model training.

## 1 Introduction

Machine learning (ML) has become a cornerstone of drug discovery, with models routinely predicting the bioactivity and safety of drug candidates in the early stages of the process [1]. Recent advancements, including foundation and generative models, hold the promise of accelerating therapeutic development across diverse modalities like small and large molecules. However, oligonucleotide therapeutics (ONTs) stand as a notable exception, with limited progress in the development of ML-based predictive models compared to small molecule and protein-based therapeutics [2].

ONTs represent a relatively new class of drugs with various mechanisms of action, including antisense oligonucleotides (ASOs), RNA interference (siRNA/miRNA/shRNA), and aptamers [3]. The first antisense drug, fomivirsen, was approved by the US FDA in 1998 for cytomegalovirus (CMV) retinitis [4]. By September 2024, 22 ONTs had received approval from the FDA or EMA, comprising 13 ASOs, 7 siRNAs, and 2 aptamers. ASOs and siRNAs are hybridization-based ONTs that function through base-pair complementarity to target mRNA transcripts. Their on-target hybridization can modulate gene expression via mechanisms such as steric blockage or RNase H/RISC-mediated cleavage. Designing these compounds involves selecting appropriate binding sites on the target transcript, which dictates their nucleobase sequences. Furthermore, chemical modifications to the ribose, phosphate, and nucleobase monomers can be introduced to enhance potency, safety, stability, and delivery [4].

---

[*]These authors contributed equally; randomized order.

39th Conference on Neural Information Processing Systems (NeurIPS 2025) Track on Datasets and Benchmarks.

The complex design landscape of oligonucleotide sequences and chemical modification patterns results in a multifaceted and laborious optimization process, underscoring the need for specialized, domain-specific ML models to expedite discovery [5]. While a seminal work in 2005 demonstrated the use of artificial neural networks to predict the inhibitory activity of unmodified siRNAs [6], the subsequent development of ML models for ONTs has been limited compared to the rapid development of models for small molecules or protein therapeutics. Moreover, most existing models for ONTs focus on predicting the efficacy of unmodified siRNAs or ASOs [6–9], with fewer models addressing the efficacy of chemically modified oligos or toxicity prediction [10–12], despite the latter being a significant hurdle in ONT development and a major contributor to clinical trial failures [13, 5]. A primary reason for the scarcity of ML models for ONTs is the lack of standardized datasets and evaluation frameworks for training and comparing new models. While small molecule property prediction has benefited from established datasets such as MoleculeNet [14] and Therapeutics Data Commons (TDC) [15], which have become benchmarks for model development, no equivalent resources exist for oligonucleotide therapeutics.

To address this critical gap, we introduce the first curated, standardized ML datasets for oligonucleotide therapeutics, encompassing a broad spectrum of modalities, therapeutically relevant properties, and targets. We employ the industry-standard HELM notation for unified molecular representation and meticulously collect and store all important metadata for each dataset [16]. Additionally, we conduct extensive benchmarking of various classical and deep learning-based models, coupled with diverse featurization and data splitting strategies, to provide a thorough evaluation of different approaches for training and assessing models for oligonucleotide drugs.

This work directly addresses a significant deficiency in the oligonucleotide therapeutics field by providing a comprehensive resource for the development and evaluation of machine learning models. By establishing standardized datasets and rigorous benchmarking protocols, our aim is to accelerate the discovery and optimization of oligonucleotide drugs, ultimately leading to improved therapeutic outcomes.

## 2 Previous and related work

### 2.1 Benchmarks in small molecule, protein and RNA therapeutics

For small molecule drug discovery, benchmark collections like MoleculeNet [14] and Therapeutics Data Commons (TDC) [15] aggregate diverse datasets covering various physicochemical properties, bioactivities, and ADMET (Absorption, Distribution, Metabolism, Excretion, Toxicity) endpoints, providing standardized data splits and evaluation metrics. Similarly, in the protein space, benchmarks such as TAPE (Tasks Assessing Protein Embeddings) [17] and ProteinGym [18] offer curated tasks and datasets derived from large protein sequence databases (e.g., Pfam, UniProt) and experimental deep mutational scans to evaluate protein representations and models on tasks relevant to protein engineering and fitness prediction. In the RNA domain, RNAGym [19] and BEACON [20] collect a similarly well-curated set of labeled datasets for RNA fitness and structural prediction.

### 2.2 Existing resources for oligonucleotide drug discovery

Databases like siRNAmod [21] and siRNAEfficacyDB [22] aggregate experimental data focused on siRNA inhibitory activity. The AOBase [23] served as a repository for antisense oligonucleotide (ASO) experimental results, collecting data from literature regarding ASO design and activity, but it is no longer maintained.

### 2.3 Limitations and gaps

Therapeutics Data Commons lacks oligonucleotide-related tasks in its otherwise extensive drug discovery benchmarks for machine learning. Current RNA benchmarks like BEACON and RNAGym concentrate on natural RNA function and structure, which are not directly relevant for oligonucleotide therapeutic endpoints. Although oligonucleotide databases like siRNAmod exist, they are limited to siRNA efficacy prediction, with no resources available for toxicity endpoints. Moreover, these resources provide datasets without standardized evaluation and benchmarking for machine learning models.

# 3 Curated oligonucleotide property datasets

## 3.1 Overview

OligoGym's dataset collection is built upon the pre-clinical oligonucleotide drug discovery pipeline, relying on various efficacy and safety benchmarks to advance molecules from initial screening to potential clinical trials. We identify on-target efficacy and toxicity profiles as crucial therapeutic endpoints influencing decisions across these stages [10, 13]. On-target efficacy serves as the primary endpoint during hit identification, while *in vitro* toxicity profiling, including general cytotoxicity and immunomodulation, is employed to de-risk candidate molecules and reflect multifaceted safety hurdles [13]. Rather than concentrating on a single property type, OligoGym offers a diverse dataset collection designed to benchmark machine learning (ML) models capable of assessing the overall developability of oligonucleotide therapeutic candidates, specifically focusing on chemically-modified oligonucleotides due to the lack of models handling these modifications in the literature. OligoGym aims to provide a training ground and assessment criteria for ML models that can significantly impact the multi-parameter optimization landscape of the oligonucleotide therapeutics discovery pipeline.

To create this collection, a review of literature and patents was conducted to find publicly accessible datasets relevant to oligonucleotide therapeutic outcomes, with an initial focus on *in vitro* assays where larger datasets are more common. Datasets previously used for predictive model development were prioritized to ensure their suitability for machine learning and allow for future comparisons with existing models. The majority of the collected data consists of efficacy endpoints, where the variable to predict is the percentage reduction or inhibition of the target gene transcript. This bias towards efficacy data is explained by the greater public availability of single- or dual-dose knockdown experiments, typically the initial assays in oligonucleotide lead identification. Conversely, toxicity or safety datasets are less common due to the higher experimental costs, which also results in *in vitro* safety evaluations being performed on a select few top compounds after efficacy screening.

We provide datasets for both antisense oligonucleotide (ASO) and short interfering RNA (siRNA) modalities. Although both are oligonucleotides, these are fundamentally different therapeutic modalities that are not biochemically interchangeable due to their distinct modes of action, structure, and chemical modifications. ASOs are single-stranded molecules that typically recruit RNase H or cause steric hindrance. Their design often incorporates a "gapmer" architecture, as exemplified in the ASOptimizer, Cytotox LNA, and TLR7/8 datasets, featuring chemically modified flank regions (e.g., LNA or MOE) to increase stability and binding affinity, flanking a central DNA "gap" to trigger RNase H-mediated degradation of the target mRNA. In contrast, siRNAs are double-stranded molecules, composed of a guide and passenger strand, that are processed by the RISC pathway for mRNA target cleavage. Their modifications, such as the 2'-fluoro and 2'-O-methyl sugars detailed in the Shmushkovich and siRNAmod datasets, are optimized for RISC loading, stability, and in some cases, unassisted cellular uptake, which presents a completely different chemical and structural optimization problem compared to ASO. We also provide a dataset from short hairpin RNA (shRNA), which are single stranded RNA folded into a stem-loop structure. shRNA differ from siRNAs in the delivery method and initial processing by DICER enzyme but otherwise utilize the same RISC pathway for gene silencing as siRNA. OligoGym provides the necessary codebase for the development of tailored predictive models for each specific modality.

## 3.2 Curation

Twelve diverse datasets (Tab. 1) encompassing regression tasks for ASO, siRNA, and shRNA efficacy and toxicity were initially curated. Each dataset underwent individual review to confirm complete compound structure information, including modifications. Reports detailing compound diversity, label distribution, and machine learning suitability were generated for each dataset. All compounds were converted to HELM notation, an industry standard for macromolecules, with any missing monomer information marked as unknown to minimize making potentially incorrect assumptions. Comprehensive details on sequence and modification patterns were provided, utilizing the HELM-CoreLibrary as the reference library for monomer identification and symbols, as recommended by the Pistoia Alliance [16]. The selected datasets cover a range of therapeutic endpoints, including efficacy, cytotoxicity, neurotoxicity, and immunomodulation. For siRNA datasets, the presence of both antisense (guide) and sense (passenger) strands was verified. Labels were either maintained as originally reported or minimally transformed to improve machine learning model training (detailed

Table 1: Summary of each dataset provided in this study. All datasets are regression tasks.

| Dataset | Modality | # Measurements | # Compounds | # Targets | Endpoints |
|---|---|---|---|---|---|
| OpenASO [25] | ASO | 3913 | 3868 | 86 | efficacy |
| ASOptimizer [9] | ASO | 32602 | 20749 | 18 | efficacy |
| TLR7 [26] | ASO | 192 | 192 | 4 | immunomodulation |
| TLR8 [26] | ASO | 192 | 192 | 4 | immunomodulation |
| Cytotox LNA [12] | ASO | 768 | 768 | 1 | cytotoxicity |
| Neurotox LNA [10] | ASO | 1825 | 1812 | 3 | neurotoxicity |
| Neurotox MOE [27–37] | ASO | 2437 | 2398 | 13 | neurotoxicity |
| siRNAmod [38, 21] | siRNA | 907 | 823 | unavailable | efficacy |
| Sherwood [39] | shRNA | 291551 | 239845 | 17802 | efficacy |
| Ichihara [40] | siRNA | 419 | 419 | 12 | efficacy |
| Huesken [6, 40] | siRNA | 2431 | 2431 | 30 | efficacy |
| Shmushkovich [41] | siRNA | 356 | 356 | unavailable | efficacy |

in Tab. A1). Additionally, natural base sequences and SMILES representations [24] were included for all compounds to allow for more detailed featurization and modeling of chemically modified oligonucleotides in the future. We contend that providing SMILES representations for oligonucleotide benchmarking datasets is crucial to prevent ambiguities in the chemical structure caused by monomer misannotation. Finally, when available, identifiers of the target mRNAs were included for each dataset.

### 3.3 Availability

The datasets are publicly accessible through our GitHub repository, which includes comprehensive code libraries enabling programmatic interaction and seamless data manipulation. This repository hosts the datasets in a standardized and user-friendly format, ensuring easy retrieval and utilization. Each dataset is accompanied by detailed metadata facilitating data understanding. Furthermore, our provided codebase offers pre-built functionalities for immediate modeling of these datasets, supporting both traditional machine learning algorithms and deep learning approaches. This allows users to quickly begin analyzing and extracting insights without the need for extensive preprocessing or model implementation. The code is available at `github.com/Roche/oligogym`.

## 4 Evaluation framework

### 4.1 Featurization

Oligonucleotide compounds are defined by their nucleobase sequences and the specific patterns of chemical modifications on the nucleobase, ribose, and phosphate groups. These modifications are critical determinants of the properties of a compound, influencing its efficacy, safety, and biodistribution. Consequently, incorporating these modifications as features could be essential for building accurate predictive models of oligonucleotide properties. In this work, we employed two distinct and modular featurization techniques, allowing for the inclusion or exclusion of chemical modification features depending on the experimental requirements.

To create features from HELM representations, we developed a simple but powerful string parser that converts a HELM string into a more human-readable format we call XNA. This intermediate format is the input to the featurizers we implemented for this work. This string parsing module is also made available with the code and can be used for a variety of tasks beyond the scope of this study.

### 4.1.1 $K$-mers featurizer

In biological sequence analysis, $k$-mers are contiguous subsequences of length $k$ extracted from a larger sequence. The composition and frequency of these $k$-mers provide a means to analyze and characterize biological sequences. Despite their straightforward nature, $k$-mers are widely employed in various analytical and predictive tasks in computational biology and remain a popular feature engineering technique for machine learning models [42]. Our $k$-mer featurizer implementation enables the selection of specific $k$ values or a range of $k$ values. For a given set of compounds, it generates a vector representation for each compound based on the $k$-mer frequencies of its nucleobase

sequence. To incorporate information about chemical modifications, we have included an option to count the occurrences of each modified monomer and append these counts to the $k$-mer frequency vector. For double-stranded compounds, the counts of $k$-mers and, if included, modified monomers are determined by considering both strands and adding them together.

### 4.1.2 One-hot encoder

One-hot encoding is a fundamental and widely adopted featurization technique for categorical data, enabling its representation as vectors. Importantly, unlike $k$-mers, one-hot encoded representations preserve positional information, and can be used for sequence-based modeling. In bioinformatics, one-hot encoding is commonly applied to biological sequences like DNA or RNA, transforming a nucleotide string of length $L$ into an $L \times 4$ matrix where each position is represented by a one-hot vector ($A = [1, 0, 0, 0]$, $C = [0, 1, 0, 0]$, $G = [0, 0, 1, 0]$, $T = [0, 0, 0, 1]$) along each position:

$$X_{base_{OHE}} \in \mathbb{R}^{L \times 4} \tag{1}$$

To achieve a comprehensive representation of oligonucleotide compounds, including their chemical modifications, we extend one-hot encoding beyond the nucleobase sequence to encompass sugar and phosphate monomers. Each unique sugar monomer in the dataset is represented by a distinct one-hot encoded vector, resulting in an $L \times S$ matrix:

$$X_{sugar_{OHE}} \in \mathbb{R}^{L \times S} \tag{2}$$

where S is the number of unique sugar monomers present in the dataset. Similarly for phosphate monomers, we obtain an $L \times P$ matrix:

$$X_{phosphate_{OHE}} \in \mathbb{R}^{L \times P} \tag{3}$$

where P is the number of unique phosphate monomers present in the dataset. These modular representations are then concatenated to form a complete feature matrix:

$$X_{OHE} = [X_{base_{OHE}}, X_{sugar_{OHE}}, X_{phosphate_{OHE}}] \in \mathbb{R}^{L \times (4+S+P)} \tag{4}$$

Note that while the individual components are one-hot encoded, the resulting $X_{OHE}$ matrix at each position will contain multiple '1's, specifically one for the base, one for the sugar, and one for the phosphate monomer, reflecting the composition of the compound at that position. In the case of double-stranded compounds, we opted to one-hot encode the two strands separately and then concatenate them along the length axis.

### 4.2 Models

For this benchmarking study, we aimed to evaluate the performance of various machine learning models on oligonucleotide property prediction tasks. Our selection included both established statistical learning algorithms and deep learning architectures to provide a comprehensive comparison. These models were chosen with the goal of establishing simple yet robust baselines, leveraging well-established architectures, several of which have also been widely applied to specific tasks within this benchmark [10, 43]. Our evaluation framework further incorporates domain knowledge by means of the tailored featurization techniques described in the previous sections.

**Linear Model**   We utilized the *scikit-learn* [44] implementation of linear regression (*LinearRegression*) and Ridge regression (*Ridge*) [45].

**Nearest Neighbors Model (KNN)**   We employed the scikit-learn *KNeighborsRegressor*, a $k$-NN regression model [46]. The number of neighbors ($k$) was treated as a hyperparameter.

**Random Forest Model (RF)**   We used the scikit-learn *RandomForestRegressor* for random forest regression [47], exploring the maximum depth and number of estimators as hyperparameters.

**XGBoost (XGB)**   We integrated the XGBoost algorithm [48] via its Python package, tuning the maximum depth and number of estimators.

**Multilayer Perceptron (MLP)**   We implemented a multi-layer perceptron architecture [49] using PyTorch Lightning, with hyperparameters encompassing the number and size of hidden layers, as well as the dropout rate.

**Convolutional Neural Network (CNN)**   We implemented a simple convolutional neural network architecture [50] in PyTorch Lightning [51]. Hyperparameters included the network depth (number of convolutional and pooling layers), the number of filters, the kernel width, and the pooling operation.

**Gated Recurrent Unit (GRU)**   We implemented a gated recurrent unit [52] architecture in PyTorch Lightning, varying its hidden dimensionality and the number of layers as hyperparameters.

**Transformer**   We implemented a simple transformer model [53] in Pytorch Lightning. The embedding dimensionality, number of attention heads, dimensionality of the feedforward network, and dropout rate were defined as hyperparameters.

## 4.3   Splitting strategies

We used two data splitting strategies to assess model performance. The first was a standard 5-fold cross-validation using random splits. Random splitting is a simple and generally applicable technique but may not capture temporal or experimental distribution shifts, potentially leading to an overly optimistic performance evaluation.

To mitigate this issue, more rigorous techniques like time-based splits (testing on more recent compounds) and molecular scaffold-based splits (grouping by chemical structure) are common in cheminformatics and QSAR modeling. In this work, we used a nucleobase splitting strategy where a $k$-mer representation of the compounds is used to perform $k$-means clustering. The cluster labels were then used to stratify the final train/test split. To evaluate the variability in model performance, similar to cross-validation, we repeated this splitting process five times with different random seeds for the clustering. While this generates five distinct splits, it does not ensure that every compound is included in a test set at least once. Random splits and nucleobase splits were performed independently for each run.

## 4.4   Experimental setup

To comprehensively benchmark feature engineering and modeling choices, an extensive experiment was conducted across a grid of featurizer and model hyperparameters. Featurizer options aimed to evaluate the impact of including chemical modification information and the choice of $k$ for the $k$-mer featurizer (Tab. A2). For each model, hyperparameters were tuned by assessing performance across a range of values (Tab. A3). A configuration comprised a specific combination of featurizer and model options.

All possible configurations were evaluated for each dataset. $K$-mer features were excluded when using sequence-based models (CNN, GRU, Transformer). Conversely, one-hot encoded features were utilized across all model architectures. For non-sequence-based models, the one-hot encoded representation was flattened into a vector. Any configuration that presented potential dimensionality issues (for example, a CNN with an excessively large kernel size relative to the input sequence length, which could lead to invalid convolution operations) was preemptively excluded from the experimental runs. Each valid configuration was evaluated with 5-fold random cross-validation or nucleobase splits as detailed in the previous section.

Neural network models were trained using the Adam optimization algorithm for a maximum of 100 epochs [54]. We used early stopping with a patience of five epochs to avoid overfitting. For every completed run configuration and across each split generated by the two splitting strategies, a suite of standard regression metrics was tracked. These metrics included the Pearson correlation coefficient (PCC), which measures the linear relationship between predicted and true values; the Spearman correlation coefficient (SCC), assessing the monotonic relationship; the mean absolute error (MAE), representing the average magnitude of errors; the root mean squared error (RMSE),

giving a higher weight to large errors; and the $R^2$ score, indicating the proportion of variance in the dependent variable that is predictable from the independent variables. This comprehensive set of metrics allowed for a thorough comparison of the performance of different feature engineering and modeling approaches.

### 4.5   Computational resources

All runs with statistical learning models (Linear, KNN, RF, XGB) were performed on CPU nodes with an Intel® Xeon® Platinum 9242 Processor, while runs with deep learning models (MLP, CNN, GRU, Transformer) were performed on GPU nodes using either a NVIDIA A100 GPU paired with an AMD EPYC 7542 CPU or a NVIDIA L40S GPU paired with an AMD EPYC 7643 CPU. Eight CPU cores were requested for each job. Jobs were parallelized and managed using a job scheduler on our in-house compute cluster.

The final benchmarking experiment consisted of 10,657 runs, where each run evaluated the combination of a featurizer and its parameters, a model class and its hyperparameters, one of two splitting strategies and the dataset to train on. The total CPU time of the experiment amounted to roughly 70 days, with an average RAM usage of 3.43 GB. The total GPU time amounted to approximately 33 days.

## 5   Results

### 5.1   Model benchmarks

To provide an overview of model performance across different datasets, we first identified the optimal feature engineering and model hyperparameter settings for each model on each dataset. This selection process was based on the results obtained using a standard 5-fold cross-validation splitting strategy. Specifically, we ranked the different configurations based on their average Pearson correlation coefficient (PCC) across the five folds. Tab. 2 and Fig. A1 present the mean PCC and the standard deviation of the PCC across the five cross-validation folds for these top-performing configurations.

The benchmark results indicate that random forest models generally performed well, emerging as the top-performing approach for 8 out of the 12 datasets. Even when not the absolute best, random forests consistently achieved competitive results, confirming previous findings that such models can deliver state-of-the-art performance on small datasets [55]. In contrast, deep learning models exhibited more variability in their performance across the datasets. Convolutional neural networks (CNNs) achieved the highest performance on three datasets, but their performance was considerably worse than traditional machine learning algorithms on others. Gated recurrent units (GRUs) showed relatively high standard deviations in performance across the cross-validation folds and generally worse results, suggesting potential difficulties in effectively training these recurrent architectures for this type of data. Despite their success in other biological datasets, transformer models did not outperform CNN and classical ML models in our benchmarks. This is likely due to the short length (under 30 nucleotides) of oligonucleotide sequences, which do not benefit from a transformer's capacity for long-range dependencies. Additionally, small sample sizes in most datasets could lead to overfitting with a complex transformer architecture.

Interestingly, models trained on tabular representations of the data (Linear, KNN, RF, XGB, MLP) either outperformed or were on par with sequence-based models (CNN, GRU, Transformer), suggesting that the association between compound features (e.g., nucleotides, chemical modifications) and measured endpoints may not strongly depend on their positional context within a sequence. We also benchmark graph neural networks (GNN), an architecture popular with small-molecule datasets, but this did not yield improved performance (Tab. A13). Details and performances of the GNN models can be found in the appendix section A.4. Furthermore, we tested a transfer learning approach of training a CNN model on embeddings from an RNA foundation model, the details can be found in the appendix section A.5.

Certain datasets, such as Shmushkovich, Ichihara, and OpenASO appeared to be intrinsically more challenging to model. A potential explanation for the Shmushkovich dataset could be the partially missing monomer information which might hinder model performance, while the challenge with the other two could be ascribed to the fact that they were gathered from different sources, introducing more experimental variability. The same analysis displaying Spearman correlation coefficient (SCC)

Table 2: Summary of model performance evaluated with a random cross-validation approach. The results of the best performing configuration for each model class are shown. Each cell contains the mean and standard deviation of the PCC across folds.

| Model Dataset | Linear | KNN | RF | XGB | MLP | CNN | GRU | Transformer |
|---|---|---|---|---|---|---|---|---|
| OpenASO | 0.33 ± 0.02 | 0.32 ± 0.01 | **0.35 ± 0.04** | 0.27 ± 0.02 | 0.27 ± 0.03 | 0.30 ± 0.03 | 0.18 ± 0.02 | 0.29 ± 0.03 |
| ASOptimizer | 0.47 ± 0.01 | 0.63 ± 0.01 | **0.64 ± 0.01** | **0.64 ± 0.01** | 0.58 ± 0.01 | 0.56 ± 0.02 | 0.57 ± 0.01 | 0.52 ± 0.02 |
| TLR7 | 0.77 ± 0.02 | 0.60 ± 0.09 | **0.78 ± 0.04** | 0.74 ± 0.07 | 0.72 ± 0.08 | 0.66 ± 0.09 | 0.35 ± 0.22 | 0.76 ± 0.04 |
| TLR8 | 0.54 ± 0.13 | 0.42 ± 0.14 | **0.68 ± 0.08** | 0.59 ± 0.14 | 0.53 ± 0.12 | 0.19 ± 0.07 | 0.16 ± 0.16 | 0.25 ± 0.19 |
| Cytotox LNA | 0.79 ± 0.02 | 0.88 ± 0.03 | **0.91 ± 0.02** | 0.91 ± 0.01 | 0.86 ± 0.01 | 0.88 ± 0.02 | 0.49 ± 0.05 | 0.73 ± 0.04 |
| Neurotox LNA | 0.64 ± 0.03 | 0.68 ± 0.03 | **0.76 ± 0.02** | 0.71 ± 0.03 | 0.63 ± 0.03 | 0.67 ± 0.04 | 0.42 ± 0.21 | 0.60 ± 0.05 |
| Neurotox MOE | 0.70 ± 0.02 | 0.69 ± 0.01 | 0.74 ± 0.02 | 0.73 ± 0.02 | 0.73 ± 0.04 | **0.75 ± 0.02** | 0.73 ± 0.02 | **0.75 ± 0.02** |
| siRNAmod | 0.62 ± 0.06 | 0.56 ± 0.04 | **0.69 ± 0.03** | 0.68 ± 0.03 | 0.62 ± 0.07 | 0.62 ± 0.04 | 0.17 ± 0.11 | 0.45 ± 0.06 |
| Sherwood | 0.78 ± 0.00 | 0.85 ± 0.00 | 0.92 ± 0.00 | **0.96 ± 0.00** | 0.82 ± 0.00 | 0.85 ± 0.00 | 0.13 ± 0.00 | 0.59 ± 0.06 |
| Ichihara | 0.54 ± 0.07 | 0.38 ± 0.07 | 0.49 ± 0.02 | 0.38 ± 0.08 | 0.54 ± 0.07 | **0.56 ± 0.02** | 0.18 ± 0.14 | 0.37 ± 0.05 |
| Huesken | 0.63 ± 0.02 | 0.45 ± 0.02 | 0.62 ± 0.04 | 0.53 ± 0.03 | 0.64 ± 0.01 | **0.65 ± 0.02** | 0.14 ± 0.03 | 0.52 ± 0.05 |
| Shmushkovich | 0.24 ± 0.09 | 0.38 ± 0.07 | **0.50 ± 0.08** | 0.41 ± 0.07 | 0.30 ± 0.21 | 0.27 ± 0.08 | 0.05 ± 0.06 | 0.17 ± 0.07 |

values showed largely the same trends observed with PCC (Tab. A4, Fig. A2). For this reason, and for simplicity, we report from now on only PCC values. We provide tables with the best featurizer and hyperparameter configurations for each combination of model, dataset and splitting strategies in Tables A5-A12.

## 5.2 The effect of the splitting strategy

With the goal of investigating the effect of the choice in splitting strategy we analyzed the results obtained when performing repeated nucleobase splits compared to random cross-validation (Tab.A15, Fig. A3). The first notable result is the generally lower performance obtained when using this splitting strategy. This is likely because generalizing across nucleobase-based splits presents a greater challenge for models due to the distributional shifts in sequence patterns between training and test sets. Notably, the performance difference between classical machine learning and deep learning models diminished, with the latter becoming the top performers for 6 out of 12 datasets. This suggests that deep learning models may possess a greater capacity to generalize across more significant distribution shifts. Furthermore, the performance metrics obtained from random cross-validation and nucleobase splitting exhibited a strong correlation (Fig. A4).

The choice of splitting strategy had a different impact on the performance evaluated on the various datasets. While some datasets (Shmushkovich, siRNAmod, and TLR8) showed similar performance distributions under both random cross-validation and nucleobase splitting, others (OpenASO, ASOptimizer, Neurotox LNA, and Cytotox LNA) exhibited a significant performance decrease with nucleobase splitting (Fig. A5). This suggests that accurate predictions for the latter group may heavily rely on specific sequence features that become underrepresented in the test set when splitting based on nucleobase patterns. Similar conclusions can be drawn when looking at the difference in the root mean squared error achieved by the top models for each splitting strategy (Fig. A6).

## 5.3 The effect of feature engineering choices

Given the diverse options for feature engineering profiled in our experiment, we aimed to understand the impact of featurization choices on prediction performance. First, we compared one-hot encoding of only the nucleobase sequence against encoding both the sequence and chemical modifications (sugar and phosphate monomers) (Fig. 1a). We observed sizable performance differences for the Cytotox LNA and siRNAmod datasets. This was anticipated, as these datasets were designed to study the effect of chemical modifications on the respective biological activities; consequently, excluding this information reduced model performance. Similarly, our $k$-mer featurizer can be configured to include modified monomer counts. Comparing results with and without these monomer features yielded similar trends to those observed with one-hot encoding (Fig. 1b), highlighting the importance of incorporating chemical modification information when relevant to the endpoint.

Another aspect we investigated for the $k$-mer featurizer was the effect of the choice of $k$ on model performance. For most datasets, including more $k$-mer lengths generally improved model performance

Table 3: Summary of model performance evaluated with nucleobase splits. The results of the best performing configuration for each model class are shown. Each cell contains the mean and standard deviation of the PCC across splits.

| Model Dataset | Linear | KNN | RF | XGB | MLP | CNN | GRU | Transformer |
|---|---|---|---|---|---|---|---|---|
| OpenASO | **0.29 ± 0.08** | 0.23 ± 0.06 | 0.29 ± 0.03 | 0.21 ± 0.02 | 0.23 ± 0.09 | 0.25 ± 0.08 | 0.19 ± 0.05 | 0.27 ± 0.11 |
| ASOptimizer | 0.39 ± 0.04 | 0.45 ± 0.06 | 0.48 ± 0.03 | 0.40 ± 0.06 | 0.43 ± 0.08 | **0.48 ± 0.04** | 0.41 ± 0.03 | 0.44 ± 0.08 |
| TLR7 | 0.68 ± 0.07 | 0.56 ± 0.13 | **0.73 ± 0.10** | 0.66 ± 0.07 | 0.66 ± 0.11 | 0.62 ± 0.17 | 0.32 ± 0.13 | 0.72 ± 0.11 |
| TLR8 | 0.59 ± 0.08 | 0.41 ± 0.13 | **0.70 ± 0.06** | 0.58 ± 0.16 | 0.57 ± 0.10 | 0.25 ± 0.19 | 0.13 ± 0.13 | 0.26 ± 0.20 |
| Cytotox LNA | 0.64 ± 0.16 | 0.64 ± 0.07 | 0.69 ± 0.09 | 0.64 ± 0.15 | **0.72 ± 0.11** | 0.70 ± 0.11 | 0.32 ± 0.23 | 0.72 ± 0.06 |
| Neurotox LNA | 0.56 ± 0.09 | 0.49 ± 0.09 | 0.56 ± 0.08 | 0.50 ± 0.16 | 0.53 ± 0.08 | **0.57 ± 0.08** | 0.17 ± 0.22 | 0.53 ±0.10 |
| Neurotox MOE | 0.59 ± 0.24 | 0.49 ± 0.05 | 0.61 ± 0.11 | 0.58 ± 0.04 | 0.66 ± 0.07 | 0.67 ± 0.10 | 0.67 ± 0.12 | **0.70 ± 0.06** |
| siRNAmod | 0.44 ± 0.11 | 0.50 ± 0.09 | **0.53 ± 0.09** | 0.42 ± 0.14 | 0.47 ± 0.07 | 0.49 ± 0.12 | 0.25 ± 0.09 | 0.49 ± 0.09 |
| Sherwood | 0.70 ± 0.05 | 0.73 ± 0.03 | 0.82 ± 0.03 | **0.90 ± 0.03** | 0.74 ± 0.04 | 0.74 ± 0.04 | 0.12 ± 0.10 | 0.44 ± 0.09 |
| Ichihara | 0.49 ± 0.09 | 0.38 ± 0.14 | 0.49 ± 0.06 | 0.34 ± 0.13 | **0.57 ± 0.12** | 0.48 ± 0.10 | 0.21 ± 0.10 | 0.35 ± 0.14 |
| Huesken | 0.62 ± 0.01 | 0.44 ± 0.01 | 0.59 ± 0.01 | 0.51 ± 0.03 | 0.62 ± 0.01 | **0.63 ± 0.03** | 0.16 ± 0.04 | 0.51 ± 0.09 |
| Shmushkovich | 0.20 ± 0.08 | 0.28 ± 0.16 | **0.42 ± 0.14** | 0.35 ± 0.07 | 0.35 ± 0.04 | 0.27 ± 0.10 | 0.12 ± 0.11 | 0.17 ± 0.20 |

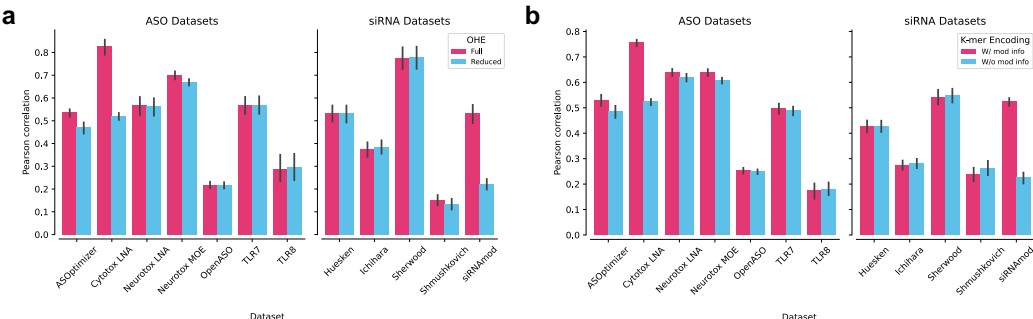

Figure 1: (a) Average PCCs across datasets of all configurations using one-hot encoding, the results obtained with a full (nucleobase and chemical modification) and a reduced (nucleobase only) encoding are compared. (b) Average PCCs across datasets of all configurations using the $k$-mer featurizer, the results obtained with and without modification information are compared. Error bars display the standard deviation.

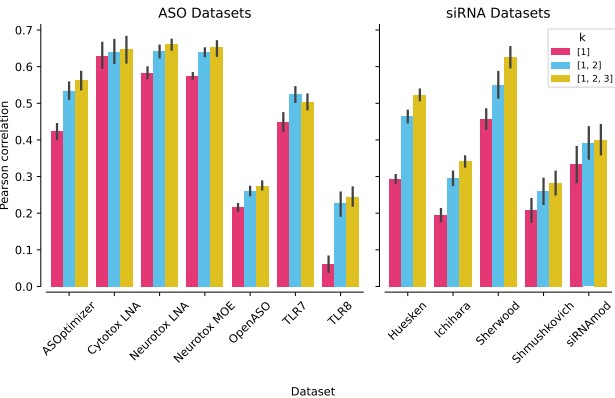

Figure 2: Average PCCs across datasets of all configurations using the $k$-mer featurizer, the results obtained with different choices of $k$ are compared. Error bars display the standard deviation.

as anticipated (Fig. 2). However, the extent of this improvement varied. While adding 3-mers resulted in only marginal gains for most datasets, a more substantial increase was observed for Sherwood, Huesken, and Ichihara. Notably, these three datasets all pertained to siRNAs, suggesting that longer sequence motifs may be more critical for determining siRNA properties compared to ASOs.

# 6  Discussion

This work introduces OligoGym, the first curated and standardized collection of datasets for oligonucleotide drug discovery, comprising 12 datasets across diverse modalities and therapeutic endpoints. It addresses a critical need for standardization in a field challenged by inconsistent molecular representations and assay readouts. We used OligoGym to benchmark a range of classical and deep learning models with tailored featurization techniques, thus establishing initial performance baselines for machine learning in oligonucleotide therapeutics.

While OligoGym represents a significant step forward, the current benchmarks do not capture the full range of possible feature engineering approaches. Notably, thermodynamic or structural features were not explored. The selection of models, though diverse, was limited to architectures suited to general regression tasks. Future work should benchmark models that incorporate domain knowledge and inductive biases. Dataset sizes are also relatively small compared to other drug modalities, often bordering on low-data regimes. This highlights the need for larger, more comprehensive datasets with sufficient technical and biological replicates to better estimate the achievable upper bounds of predictive performance. Biological variability and assay noise further underscore the value of multiple measurements per compound.

Equally important is the real-world utility of benchmarks. In oligonucleotide discovery, the performance thresholds for ML models vary with the discovery stage. During hit identification, a rank correlation greater than 0.7 can prioritize the top 10–20 % of candidates for experimental validation, typically out of 1,000–2,000 compounds [56]. For lead optimization, where precise property improvements are sought in hundreds of compounds, metrics like RMSE are more appropriate, ideally matching the assay's standard deviation for reliability [57]. For toxicity endpoints, a recall of at least 0.95 at a defined cutoff would minimize the risk of advancing unsafe compounds. Accordingly, OligoGym provides a range of performance metrics tailored to different discovery stages.

Despite its limitations, OligoGym lowers the entry barrier for ML research in oligonucleotide therapeutics by providing accessible, ML-ready datasets and well-defined challenges. Oligonucleotide drugs' reliance on chemical modifications creates a complex interplay between biological sequence and medicinal chemistry, offering unique challenges for multi-scale modeling that is relevant beyond this domain [58–60]. To facilitate research, we provide natural base, SMILES, and HELM sequences for all datasets. The scarcity and synthetic nature of these molecules limit the use of pretrained foundation models (see Appendix A.5), highlighting the importance of low-data and few-shot learning, another emerging direction in drug discovery [61]. Furthermore, the multifaceted nature of oligonucleotide development opens opportunities for research in multi-objective optimization.

# 7  Conclusion

In conclusion, OligoGym offers a foundational and standardized platform poised to catalyze the application of machine learning in oligonucleotide drug discovery. By providing a curated collection of datasets and establishing initial benchmarks, this resource lowers the barrier for computational researchers entering the field. Recognizing the existing limitations in feature exploration, model selection, and dataset sizes, future work should focus on expanding these aspects to unlock the full potential of machine learning for this therapeutic modality. Ultimately, we anticipate that OligoGym will serve as a valuable community resource, fostering collaboration and accelerating the development of novel oligonucleotide-based therapeutics through enhanced data sharing and predictive modeling.

## Acknowledgments and Disclosure of Funding

We thank Alexander Wyss for their help in reviewing the code used for the project and setting up CI/CD pipelines. We thank Dimitar Yonchev and Daniel Butnaru for their feedback on the project. We thank Shushan Toneyan for their feedback on the manuscript. We thank Peter Hagedorn for their initial gathering of the Neurotox MOE dataset.

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

# A Appendix

## A.1 Description of curated datasets

In addition to the curation process described in the main text we provide an overview of the study and dataset generation from the original sources below and summarize them in Tab.A1.

Table A1: Description of each dataset and the labels extracted from the original source.

| Name | Description |
|---|---|
| OpenASO [25] | An ASO efficacy dataset collected by IDT from an ASO database and also from patent data. According to the publication each ASO has a full phosphorothioate backbone and no chimeric sequences. The efficacy is reported as an activity level range from 0 (complete target inhibition) to 1 (no difference in target activity when compared to appropriate control). The activity was measured from either the protein product or direct mRNA abundance. The dataset is obtained from a pre-curated GitHub repository.`https://github.com/lackeylela/openASO`. |
| ASOptimizer [9] | A collection of inhibitory activity for different gapmer ASOs collected from various patents and publication. The labels are reported as percentage inhibition of target mRNA relative to untreated controls with concentration of the ASO also provided. |
| siRNAmod [38, 21] | A curated siRNA efficacy dataset that contains only sugar and base information. The labels are reported as percentage knockdown of the target mRNA. |
| Sherwood [39] | A biased screening of shRNAs using the miR-30 scaffold. The DSIR algorithm was used to select for sequences likely to be active. The labels are reported as minmax-scaled scores. The raw scores were obtained from a high-throughput assay based on a SORT-seq strategy using a fluorescence reporter construct with the target site in the 3' UTR of the reporter gene. |
| Shmushkovich [41] | A study of cholesterol-conjugated siRNA efficacy using dual luciferase reporter in HeLa cells at 1 $\mu$M. The labels are reported as percentage remaining relative to a control siRNA. |
| Ichihara [40] | A pre-curated collection of unmodified siRNA efficacy data collected from five different studies with different assays. The labels are reported as percentage inhibition relative to control from various assays as reported by authors of the curated dataset. |
| Huesken [6, 40] | A dataset of unmodified siRNA efficacy for various target genes. The labels are reported as percentage inhibition relative to a control using an eCFP-eYFP dual reporter assay. |
| TLR8 [26] | 2'OMe gapmer screen of TLR8 potentiation. The labels are reported as TLR8 levels after induction with 100 $n$M ASO as measured by fluorescence as using NF-$\kappa$B-Luciferase reporter in HEK293-TLR8 cells. The values provided are relative to those obtained after induction with Resiquimod (= 1.0). |
| TLR7 [26] | 2'OMe gapmer screen of TLR7 inhibition. The labels are reported as TLR7 levels after induction with 100 $n$M ASO as measured by fluorescence as using NF-$\kappa$B-Luciferase reporter in HEK293-TLR7 cells. The values provided are relative to those obtained after induction with Resiquimod (=100). |
| Cytotox LNA [12] | A study looking at the relationship between nucleobase and LNA sequence on ASO activity and cytotoxicity. The labels are reported as minmax-scaled average Caspase levels (N=3) measured in HeLa cells. |
| Neurotox MOE [27–37] | Acute neurotoxicity data for MOE modifed ASOs scraped from patents. The labels are rounded 3-hour Functional Observational Battery (FOB) scores in mice (7 categories: 0 (safe) to 7(toxic)). |
| Neurotox LNA [10] | Acute neurotoxicity of LNA gapmers as measured with Calcium oscillation scores in neuronal cells. |

### A.1.1 OpenASO

The OpenASO dataset originates from a study aimed at improving the rational design of antisense oligonucleotides (ASOs) by identifying short sequence motifs (2-5 bases) significantly associated with high or low gene suppression activity [25]. The goal was to discover sequence features beyond simple thermodynamics that influence efficacy, thereby enhancing predictive models. The dataset comprises 3913 ASO sequences compiled from the ODNBase database and USPTO patent data. Strict inclusion criteria were applied: sequences had to possess a complete phosphorothioate backbone, excluding chimeric molecules. The associated activity levels represent experimentally measured gene suppression at the cellular level (via mRNA or protein quantification) on a normalized scale from 0.0 (complete inhibition) to 1.0 (no inhibition). Because the original dataset is no longer available, the dataset was instead obtained from a pre-curated GitHub repository (`https://github.com/lackeylela/openASO`).

### A.1.2 ASOptimizer

The ASOptimizer dataset was collected for the study by Hwang et al. [9], which aimed to develop a deep-learning framework, named ASOptimizer, for the efficient design of potent and safe RNase H-mediated antisense oligonucleotides (ASOs). The framework focuses on both selecting optimal ASO target sites on mRNA and optimizing ASO chemical modification patterns to enhance inhibitory activity and reduce cytotoxicity. The data collection process involved compiling a large database of 187,090 experimental results from granted patents (sourced via Lens.org) and scientific publications. This extensive effort required manually extracting ASO sequences, details of their chemical modifications, experimental conditions (cell line, dose, transfection method, etc.), and the corresponding target mRNA inhibition rates, typically measured by qRT-PCR. While the full dataset was not publicly available, the authors released a subset of 32,602 datapoints used in original model development; this is the subset curated here.

### A.1.3 siRNAmod

The siRNAmod dataset used in [38] was derived from the siRNAmod database, a specialized repository created to address the lack of centralized resources for chemically modified siRNAs (cm-siRNAs) [21]. The aim of the database was to consolidate scattered experimental data to facilitate research and therapeutic development. They manually curated the database by exhaustively searching literature (initially screening 900 PubMed articles), extracting information for 4894 experimentally validated cm-siRNAs from 96 articles. The curated data included siRNA sequences, details on 128 unique chemical modifications (type, position, structure, SMILES), measured biological efficacy (e.g., percentage knockdown), target gene, cell line, experimental assay details, and references. In [38] the authors subsequently utilized this database, applying filtering criteria (e.g., 21-mer length, specific activity reporting, minimum modification frequency) to obtain a subset of 907 chemically modified siRNAs for training and evaluating machine learning models to predict siRNA silencing efficiency based on sequence and modification patterns.

### A.1.4 Sherwood

The Sherwood dataset was generated with the primary aim of developing "shERWOOD," a computational algorithm to accurately predict the gene knockdown potency of short hairpin RNAs (shRNAs), addressing the previous lack of reliable shRNA design tools [39]. To create the extensive dataset needed for algorithm training, they employed a high-throughput, multiplexed sensor assay to measure the efficacy of a large pool of shRNAs. This involved synthesizing libraries of doxycycline-inducible shRNA constructs, each paired with a GFP-tagged target sequence, which were then introduced into reporter cells. Upon shRNA expression, potent shRNAs led to a reduction in GFP signal; cells exhibiting low GFP were isolated via fluorescence-activated cell sorting (FACS), and the enriched shRNA sequences were identified and quantified using next-generation sequencing. The resulting efficacy data, derived from both unbiased shRNA tiling libraries and shRNAs pre-selected by the DSIR algorithm, were consolidated into single potency scores for each shRNA to train the shERWOOD prediction model.

### A.1.5 Shmushkovich

The Shmushkovich et al. [41] study aimed to identify key functional features determining the efficacy of heavily chemically modified, cholesterol-conjugated, self-deliverable siRNAs (sdRNAs) and to develop a predictive algorithm specifically for such modified oligonucleotides, as existing models for unmodified siRNAs proved unsuitable. The dataset was generated by synthesizing a panel of 356 unique sdRNAs targeting different genes, all featuring extensive chemical modification pattern including 2'-fluoro and 2'-O-methyl sugars, phosphorothioate linkages, and a 3'-cholesterol conjugate on the sense strand for unassisted cellular uptake. The efficacy of these sdRNAs was evaluated using a dual-luciferase reporter system in HeLa cells; target sequences were cloned into the 3' UTR of Renilla luciferase in psiCheck-2 vectors, and cells were treated with 1 μM of each sdRNA, with knockdown quantified as the percentage of Renilla luciferase expression remaining relative to controls after 48 hours.

### A.1.6 Ichihara

The Ichihara dataset, as described by Ichihara et al. [40], was primarily compiled to validate their siRNA activity prediction algorithm, "i-Score." The study aimed to create a simple yet effective algorithm based exclusively on nucleotide preferences at each of the 19 positions in an siRNA sequence, using a linear regression model. The dataset, referred to as "Dataset B" in their publication, consists of 419 unmodified 19-mer siRNA sequences and their corresponding experimentally determined inhibitory activities, which were manually curated from five previously published research articles. This collection served as an independent validation set to assess the predictive accuracy of i-Score and to investigate the influence of siRNA duplex thermostability on prediction dependability.

### A.1.7 Huesken

The Huesken et al. [6] study aimed to develop a robust algorithm, BIOPREDsi, using an artificial neural network (ANN) to predict the efficacy of small interfering RNAs (siRNAs), which would then be used to design a genome-wide siRNA library with a high likelihood of potent gene knockdown. To generate the large, homogeneous dataset required for training the ANN, they screened 3,106 randomly selected 21-mer siRNAs (typically with dTdT overhangs) targeting 34 different mRNA species (including human and rodent genes) using a high-throughput dual-fluorescent reporter gene system (eYFP as target, eCFP as control) in H1299 cells. After applying quality control filters, a final dataset of 2,431 siRNAs with their corresponding inhibition percentages (derived from the eYFP/eCFP ratio) was established; this dataset formed the basis for training (2,182 siRNAs) and testing (249 siRNAs) the BIOPREDsi algorithm.

### A.1.8 TLR8 and TLR7

The TLR7 and TLR8 datasets described by Alharbi et al. [26] were generated to investigate the sequence-dependent immunomodulatory effects of 2'-O-methyl (2'OMe) gapmer antisense oligonucleotides (ASOs) on Toll-Like Receptor 7 (TLR7) and TLR8 signaling. The study aimed to characterize ASO structural and sequence determinants that cause divergent outcomes—specifically, suppression of TLR7 versus potentiation of TLR8 activity—and to enable the rational design of ASOs that can selectively modulate these receptors, for instance, to enhance TLR8-mediated immune responses for immunotherapy while avoiding TLR7 suppression. The dataset comprises activity measurements for a library of 192 unique 2'OMe gapmer ASOs (with phosphorothioate backbones, designed against four different human transcripts) which were screened for their effects on TLR7 and TLR8 signaling in HEK293 cells stably expressing either human TLR7 or TLR8, along with an NF-$\kappa$B-luciferase reporter. Cells were treated with the ASOs (100 nM or 500 nM) before stimulation with the TLR7/8 agonist Resiquimod (R848), and the resulting modulation (inhibition for TLR7, potentiation for TLR8) was quantified by changes in luciferase activity relative to R848-only controls.

### A.1.9 Cytotox LNA

The Cytotox LNA dataset, from the study by Papargyri et al. [12], was generated to systematically investigate how the chemical and structural diversity of Locked Nucleic Acid (LNA)-modified gapmer antisense oligonucleotides influences their functional properties, particularly their target knockdown activity and cytotoxic potential. The study aimed to establish structure-activity relationships to guide

the optimization of LNA gapmers for improved pharmacological profiles. To achieve this, they designed and synthesized 768 architecturally diverse, iso-sequential LNA gapmers (16-mers and 13-mers with full phosphorothioate backbones) targeting two distinct regions of the human HIF1A mRNA. These gapmers featured systematically varied numbers and positions of LNA modifications in their flank regions. The dataset relevant to "Cytotox LNA" comprises the cytotoxic potential measurements obtained by treating HeLa cells with each of these 768 gapmers (typically at 100 nM via transfection) and quantifying induced apoptosis through caspase activation assays 24 hours later.

### A.1.10 Neurotox MOE

The Neurotox MOE dataset was newly curated to aggregate data on the acute neurotoxicity of 2'-O-Methoxyethyl (MOE) modified antisense oligonucleotides (ASOs). The data collection process involved manually scraping information from numerous patents. [27–37] Specifically, Functional Observational Battery (FOB) scores from studies in mice and rats observed around 3 hours post-administration were extracted, and these scores were then rounded (e.g., into 7 categories where 0 indicates safe and >7 indicates toxic) to serve as labels for the dataset.

### A.1.11 Neurotox LNA

The Neurotox LNA dataset by Hagedorn et al. [10] was generated as part of a study aimed at predicting and mitigating acute, non-hybridization-dependent neurotoxicity of Locked Nucleic Acid (LNA)-modified antisense oligonucleotides (ASOs) following intracerebroventricular (ICV) dosing in mice. The primary goal was to develop a sequence-based bioinformatics tool to identify ASO designs with acceptable neurotoxic potential, thereby reducing the reliance on animal testing. The dataset primarily consists of *in vitro* data from LNA-modified ASOs (targeting MAPT pre-mRNA), which were evaluated for their effects on spontaneous calcium oscillations in primary rat cortical neurons using a fluorescent calcium indicator assay; this cellular assay was found to correlate with acute neurobehavioral side effects observed in mice. The study identified key sequence features (e.g., guanine and adenine content, G-free stretch from the 3'-end) that influence these calcium oscillation scores and subsequently developed a weighted linear regression model to predict this neurotoxic potential.

## A.2 Hyperparameter grid search for baseline models benchmarking

Table A2: Featurizer hyperparameters.

| Featurizer | Parameter | Values |
|---|---|---|
| K-mers | K | [1], [1, 2], [1, 2, 3] |
| | modification information | False, True |
| One-hot encoder | encoded components | [base], [base, sugar, phosphate] |

Table A3: Model hyperparameters.

| Model | Parameter | Values |
|---|---|---|
| Linear | L2 weight | 0, 1 |
| KNN | K | 5, 10, 15 |
| RF | max depth | 10, 20, 30 |
| | number of estimators | 100, 500, 1000 |
| XGB | max depth | 10, 20, 30 |
| | number of estimators | 100, 500, 1000 |
| MLP | number of hidden layers | 1, 2 |
| | hidden layer dimensionality | 64, 128 |
| | dropout rate | 0, 0.25 |
| CNN | depth | 1, 2 |
| | number of filters | 32, 64 |
| | kernel width | 3, 5, 7 |
| | pooling operation | max, avg |
| GRU | number of hidden layers | 1, 2 |
| | hidden layer dimensionality | 32, 64 |
| Transformer | embedding dimensionality | 64, 128 |
| | number of heads | 2, 4 |
| | feedforward network dimensionality | 64, 128 |
| | number of transformer layers | 1, 2 |
| | dropout rate | 0, 0.25 |

## A.3 Additional results

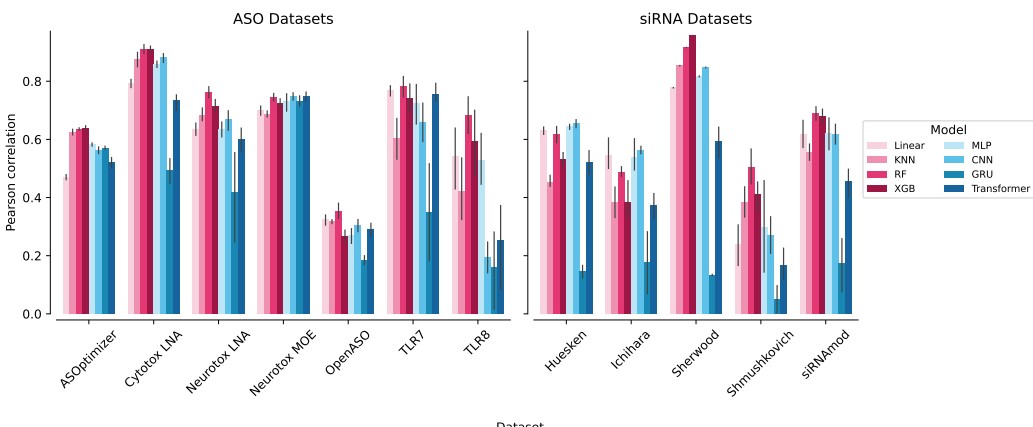

Figure A1: PCC of the best performing configurations for each model and dataset, evaluated with a random cross-validation approach. Error bars display the standard deviation.

Table A4: Summary of model performance evaluated with a random cross-validation approach. The results of the best performing configuration for each model class are shown. Each cell contains the mean and standard deviation of the SCC across folds.

| Model Dataset | Linear | KNN | RF | XGB | MLP | CNN | GRU | Transformer |
|---|---|---|---|---|---|---|---|---|
| OpenASO | 0.31 ± 0.02 | 0.29 ± 0.03 | **0.33 ± 0.03** | 0.25 ± 0.02 | 0.26 ± 0.02 | 0.30 ± 0.02 | 0.18 ± 0.03 | 0.27 ± 0.03 |
| ASOptimizer | 0.47 ± 0.01 | 0.62 ± 0.01 | **0.63 ± 0.01** | **0.63 ± 0.01** | 0.58 ± 0.01 | 0.56 ± 0.01 | 0.56 ± 0.01 | 0.51 ± 0.02 |
| TLR7 | 0.77 ± 0.02 | 0.59 ± 0.09 | **0.79 ± 0.04** | 0.73 ± 0.07 | 0.71 ± 0.10 | 0.67 ± 0.08 | 0.31 ± 0.22 | 0.74 ± 0.07 |
| TLR8 | 0.55 ± 0.14 | 0.41 ± 0.08 | **0.62 ± 0.10** | 0.57 ± 0.13 | 0.52 ± 0.12 | 0.16 ± 0.06 | 0.14 ± 0.14 | 0.25 ± 0.20 |
| Cytotox LNA | 0.83 ± 0.03 | 0.89 ± 0.03 | 0.92 ± 0.00 | **0.93 ± 0.01** | 0.88 ± 0.01 | 0.91 ± 0.01 | 0.51 ± 0.05 | 0.76 ± 0.04 |
| Neurotox LNA | 0.62 ± 0.03 | 0.64 ± 0.05 | **0.72 ± 0.01** | 0.69 ± 0.03 | 0.62 ± 0.03 | 0.64 ± 0.03 | 0.41 ± 0.22 | 0.61 ± 0.04 |
| Neurotox MOE | 0.72 ± 0.01 | 0.69 ± 0.01 | **0.76 ± 0.01** | 0.74 ± 0.02 | 0.74 ± 0.02 | 0.75 ± 0.01 | 0.74 ± 0.02 | 0.74 ± 0.01 |
| siRNAmod | 0.60 ± 0.05 | 0.55 ± 0.05 | **0.72 ± 0.03** | 0.70 ± 0.04 | 0.62 ± 0.05 | 0.59 ± 0.03 | 0.05 ± 0.08 | 0.44 ± 0.05 |
| Sherwood | 0.77 ± 0.00 | 0.83 ± 0.00 | 0.90 ± 0.00 | **0.95 ± 0.00** | 0.80 ± 0.00 | 0.83 ± 0.00 | 0.17 ± 0.01 | 0.58 ± 0.07 |
| Ichihara | **0.56 ± 0.07** | 0.42 ± 0.07 | 0.51 ± 0.05 | 0.39 ± 0.07 | 0.55 ± 0.06 | 0.54 ± 0.05 | 0.22 ± 0.14 | 0.36 ± 0.04 |
| Huesken | 0.65 ± 0.01 | 0.46 ± 0.03 | 0.62 ± 0.04 | 0.54 ± 0.04 | 0.65 ± 0.01 | **0.66 ± 0.03** | 0.15 ± 0.03 | 0.53 ± 0.06 |
| Shmushkovich | 0.23 ± 0.10 | 0.36 ± 0.07 | **0.44 ± 0.08** | 0.36 ± 0.05 | 0.28 ± 0.11 | 0.30 ± 0.07 | 0.07 ± 0.19 | 0.24 ± 0.14 |

Table A5: Best hyperparameters and featurizers for the linear model across splitting strategies. $\alpha$ is the weight of the L2 regularization term for the linear model.

| Splitting strategy Dataset | Random | Nucleobase |
|---|---|---|
| OpenASO | KMers (k=[1, 2], mod=False), $\alpha$=0 | KMers (k=[1, 2], mod=False), $\alpha$=1 |
| ASOptimizer | OHE (full), $\alpha$=1 | OHE (full), $\alpha$=1 |
| TLR7 | OHE (full), $\alpha$=1 | OHE (full), $\alpha$=1 |
| TLR8 | KMers (k=[1, 2], mod=True), $\alpha$=0 | OHE (full), $\alpha$=0 |
| Cytotox LNA | OHE (full), $\alpha$=1 | KMers (k=[1], mod=True), $\alpha$=1 |
| Neurotox LNA | KMers (k=[1, 2, 3], mod=True), $\alpha$=0 | OHE (full), $\alpha$=1 |
| Neurotox MOE | OHE (full), $\alpha$=1 | OHE (full), $\alpha$=1 |
| siRNAmod | OHE (full), $\alpha$=1 | KMers (k=[1], mod=True), $\alpha$=1 |
| Sherwood | OHE (full), $\alpha$=1 | OHE (full), $\alpha$=0 |
| Ichihara | OHE (full), $\alpha$=1 | OHE (full), $\alpha$=1 |
| Huesken | KMers (k=[1, 2, 3], mod=False), $\alpha$=1 | KMers (k=[1, 2, 3], mod=True), $\alpha$=0 |
| Shmushkovich | OHE (full), $\alpha$=1 | KMers (k=[1, 2, 3], mod=False), $\alpha$=1 |

Table A6: Best hyperparameters and featurizers for the k-nearest neighbors model across splitting strategies. K is the number of neighbors used by the model.

| Splitting strategy
Dataset | Random | Nucleobase |
|---|---|---|
| OpenASO | KMers (k=[1, 2, 3], mod=True), K=10 | KMers (k=[1, 2], mod=True), K=10 |
| ASOptimizer | KMers (k=[1, 2, 3], mod=True), K=10 | KMers (k=[1, 2, 3], mod=True), K=15 |
| TLR7 | KMers (k=[1], mod=True), K=10 | OHE (full), K=15 |
| TLR8 | OHE (full), K=15 | OHE (full), K=10 |
| Cytotox LNA | OHE (full), K=5 | OHE (full), K=15 |
| Neurotox LNA | KMers (k=[1, 2, 3], mod=True), K=5 | KMers (k=[1, 2], mod=True), K=15 |
| Neurotox MOE | KMers (k=[1, 2, 3], mod=True), K=10 | KMers (k=[1, 2, 3], mod=True), K=15 |
| siRNAmod | KMers (k=[1, 2, 3], mod=True), K=10 | KMers (k=[1], mod=True), K=10 |
| Sherwood | OHE (full), K=15 | OHE (full), K=15 |
| Ichihara | KMers (k=[1, 2], mod=False), K=5 | OHE (full), K=15 |
| Huesken | OHE (full), K=15 | OHE (full), K=15 |
| Shmushkovich | KMers (k=[1], mod=False), K=15 | KMers (k=[1, 2], mod=False), K=15 |

Table A7: Best hyperparameters and featurizers for the random forest model. D is the maximum depth of the tree and N is the number of estimators.

| Splitting strategy
Dataset | Random | Nucleobase |
|---|---|---|
| OpenASO | KMers (k=[1, 2, 3], mod=True), D=10, N=500 | KMers (k=[1, 2, 3], mod=True), D=20, N=1000 |
| ASOptimizer | KMers (k=[1, 2, 3], mod=True), D=20, N=500 | KMers (k=[1, 2, 3], mod=True), D=20, N=1000 |
| TLR7 | OHE (full), D=20, N=500 | OHE (full), D=10, N=1000 |
| TLR8 | OHE (full), D=20, N=500 | OHE (full), D=30, N=100 |
| Cytotox LNA | OHE (full), D=10, N=500 | KMers (k=[1, 2, 3], mod=True), D=30, N=1000 |
| Neurotox LNA | KMers (k=[1, 2, 3], mod=True), D=20, N=500 | KMers (k=[1, 2], mod=True), D=20, N=100 |
| Neurotox MOE | OHE (full), D=30, N=500 | OHE (full), D=10, N=100 |
| siRNAmod | OHE (full), D=10, N=500 | OHE (full), D=20, N=1000 |
| Sherwood | OHE (full), D=10, N=500 | OHE (full), D=10, N=500 |
| Ichihara | OHE (full), D=20, N=1000 | OHE (full), D=30, N=500 |
| Huesken | OHE (full), D=30, N=1000 | OHE (full), D=20, N=100 |
| Shmushkovich | KMers (k=[1, 2, 3], mod=True), D=20, N=100 | KMers (k=[1, 2], mod=False), D=10, N=500 |

Table A8: Best hyperparameters and featurizers for the XGBoost model. D is the maximum depth of the tree and N is the number of estimators.

| Splitting strategy
Dataset | Random | Nucleobase |
|---|---|---|
| OpenASO | KMers (k=[1, 2], mod=False), D=10, N=1000 | KMers (k=[1, 2, 3], mod=True), D=20, N=1000 |
| ASOptimizer | KMers (k=[1, 2, 3], mod=True), D=10, N=100 | OHE (full), D=10, N=500 |
| TLR7 | OHE (full), D=10, N=100 | OHE (full), D=10, N=500 |
| TLR8 | OHE (full), D=10, N=1000 | OHE (full), D=20, N=100 |
| Cytotox LNA | OHE (full), D=10, N=100 | KMers (k=[1], mod=True), D=10, N=1000 |
| Neurotox LNA | KMers (k=[1, 2, 3], mod=True), D=20, N=500 | KMers (k=[1, 2], mod=True), D=30, N=1000 |
| Neurotox MOE | OHE (full), D=10, N=500 | OHE (full), D=10, N=500 |
| siRNAmod | OHE (full), D=10, N=1000 | KMers (k=[1, 2], mod=True), D=20, N=500 |
| Sherwood | OHE (full), D=10, N=500 | OHE (full), D=10, N=500 |
| Ichihara | KMers (k=[1, 2, 3], mod=True), D=30, N=1000 | OHE (full), D=20, N=1000 |
| Huesken | OHE (full), D=10, N=1000 | OHE (full), D=10, N=500 |
| Shmushkovich | KMers (k=[1], mod=False), D=10, N=100 | KMers (k=[1, 2], mod=False), D=20, N=100 |

Table A9: Best hyperparameters and featurizers for the MLP model. H are the hidden layer dimensionalities and D is the dropout rate.

| Splitting strategy
Dataset | Random | Nucleobase |
|---|---|---|
| OpenASO | OHE (full), H=[128], D=0 | OHE (full), H=[64, 64], D=0 |
| ASOptimizer | KMers (k=[1, 2, 3], mod=True), H=[128, 128], D=0 | OHE (full), H=[64, 64], D=0 |
| TLR7 | OHE (full), H=[64, 64], D=0 | OHE (full), H=[64, 64], D=0 |
| TLR8 | OHE (full), H=[64], D=0 | OHE (full), H=[64], D=0 |
| Cytotox LNA | OHE (full), H=[64], D=0 | KMers (k=[1], mod=True), H=[128], D=0 |
| Neurotox LNA | OHE (full), H=[128], D=0 | OHE (full), H=[64], D=0 |
| Neurotox MOE | OHE (full), H=[128, 128], D=0 | OHE (full), H=[64], D=0 |
| siRNAmod | OHE (full), H=[128], D=0 | OHE (full), H=[64], D=0.25 |
| Sherwood | OHE (full), H=[64], D=0 | OHE (full), H=[128], D=0 |
| Ichihara | OHE (full), H=[128], D=0.25 | OHE (full), H=[128], D=0 |
| Huesken | OHE (full), H=[128], D=0 | OHE (full), H=[64], D=0 |
| Shmushkovich | KMers (k=[1], mod=False), H=[64, 64], D=0 | KMers (k=[1], mod=False), H=[128, 128], D=0 |

Table A10: Best hyperparameters and featurizers for the CNN model. H is the number of convolutional filters used, W is the kernel width of the filters, D is the depth of the model and P is the pooling operation.

| Splitting strategy
Dataset | Random | Nucleobase |
|---|---|---|
| OpenASO | OHE (full), H=32, W=5, D=1, P=max | OHE (full), H=64, W=3, D=1, P=avg |
| ASOptimizer | OHE (full), H=64, W=5, D=2, P=avg | OHE (full), H=64, W=3, D=2, P=avg |
| TLR7 | OHE (full), H=64, W=7, D=1, P=max | OHE (full), H=64, W=3, D=1, P=avg |
| TLR8 | OHE (full), H=64, W=3, D=2, P=avg | OHE (full), H=64, W=3, D=2, P=avg |
| Cytotox LNA | OHE (full), H=64, W=5, D=2, P=max | OHE (full), H=64, W=5, D=1, P=avg |
| Neurotox LNA | OHE (full), H=64, W=5, D=2, P=max | OHE (full), H=32, W=3, D=2, P=avg |
| Neurotox MOE | OHE (full), H=64, W=5, D=2, P=avg | OHE (full), H=32, W=3, D=1, P=max |
| siRNAmod | OHE (full), H=64, W=7, D=1, P=max | OHE (full), H=32, W=5, D=1, P=max |
| Sherwood | OHE (full), H=64, W=7, D=1, P=max | OHE (full), H=32, W=5, D=1, P=max |
| Ichihara | OHE (full), H=64, W=7, D=1, P=max | OHE (full), H=64, W=5, D=1, P=avg |
| Huesken | OHE (full), H=64, W=3, D=1, P=max | OHE (full), H=64, W=3, D=1, P=max |
| Shmushkovich | OHE (full), H=64, W=5, D=1, P=avg | OHE (full), H=64, W=3, D=1, P=max |

Table A11: Best hyperparameters and featurizers for the GRU model. H is the hidden dimensionality and N is the number of layers.

| Splitting strategy
Dataset | Random | Nucleobase |
|---|---|---|
| OpenASO | OHE (full), H=64, N=1 | OHE (full), H=32, N=2 |
| ASOptimizer | OHE (full), H=64, N=2 | OHE (full), H=32, N=1 |
| TLR7 | OHE (full), H=64, N=2 | OHE (full), H=64, N=2 |
| TLR8 | OHE (full), H=32, N=2 | OHE (full), H=32, N=2 |
| Cytotox LNA | OHE (full), H=32, N=2 | OHE (full), H=64, N=2 |
| Neurotox LNA | OHE (full), H=64, N=2 | OHE (full), H=32, N=1 |
| Neurotox MOE | OHE (full), H=64, N=1 | OHE (full), H=64, N=1 |
| siRNAmod | OHE (full), H=64, N=2 | OHE (full), H=64, N=2 |
| Sherwood | OHE (full), H=32, N=1 | OHE (full), H=32, N=1 |
| Ichihara | OHE (full), H=64, N=1 | OHE (full), H=32, N=2 |
| Huesken | OHE (full), H=64, N=1 | OHE (full), H=64, N=1 |
| Shmushkovich | OHE (full), H=64, N=1 | OHE (full), H=64, N=1 |

Table A12: Best hyperparameters and featurizers for the transformer model. H is the embedding dimensionality, N is the number of layers, F is the dimensionality of the feedforward network, A is the number of attention heads and D is the dropout rate.

| Splitting strategy Dataset | Random | Nucleobase |
|---|---|---|
| OpenASO | OHE (full), H=128, N=2, F=64, A=2, D=0 | OHE (full), H=128, N=1, F=64, A=2, D=0 |
| ASOptimizer | OHE (full), H=64, N=2, F=128, A=4, D=0 | OHE (full), H=128, N=1, F=64, A=4, D=0.25 |
| TLR7 | OHE (full), H=64, N=2, F=64, A=4, D=0 | OHE (full), H=64, N=2, F=128, A=4, D=0 |
| TLR8 | OHE (full), H=64, N=2, F=64, A=4, D=0 | OHE (full), H=128, N=2, F=64, A=4, D=0 |
| Cytotox LNA | OHE (full), H=128, N=2, F=64, A=4, D=0 | OHE (full), H=128, N=2, F=128, A=2, D=0.25 |
| Neurotox LNA | OHE (full), H=64, N=1, F=128, A=4, D=0 | OHE (full), H=128, N=1, F=128, A=2, D=0 |
| Neurotox MOE | OHE (full), H=64, N=1, F=64, A=2, D=0 | OHE (full), H=128, N=2, F=128, A=2, D=0.25 |
| siRNAmod | OHE (full), H=64, N=1, F=128, A=2, D=0 | OHE (full), H=64, N=1, F=64, A=2, D=0 |
| Sherwood | OHE (full), H=64, N=2, F=64, A=4, D=0 | OHE (full), H=128, N=2, F=64, A=2, D=0 |
| Ichihara | OHE (full), H=64, N=1, F=64, A=4, D=0 | OHE (full), H=128, N=1, F=64, A=4, D=0.25 |
| Huesken | OHE (full), H=128, N=1, F=128, A=2, D=0.25 | OHE (full), H=64, N=1, F=64, A=4, D=0.25 |
| Shmushkovich | OHE (full), H=128, N=2, F=128, A=2, D=0 | OHE (full), H=64, N=2, F=128, A=4, D=0 |

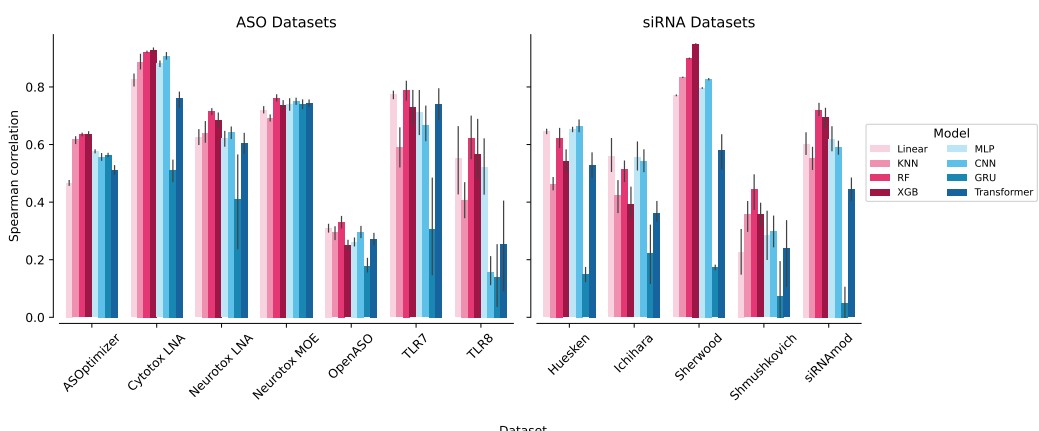

Figure A2: SCCs of the best performing configurations for each model and dataset, evaluated with a random cross-validation approach. Error bars display the standard deviation.

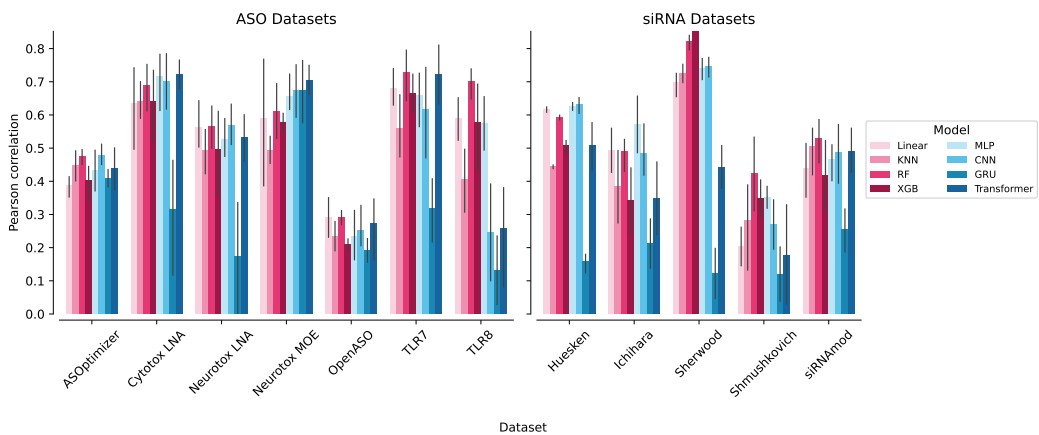

Figure A3: PCCs of the best performing configurations for each model and dataset, evaluated on nucleobase splits. Error bars display the standard deviation.

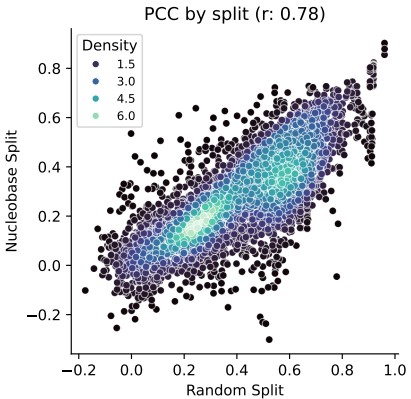

Figure A4: Average PCCs evaluated on the test set (average across folds) from each configuration measured with a random split (x-axis) versus a nucleobase split (y-axis).

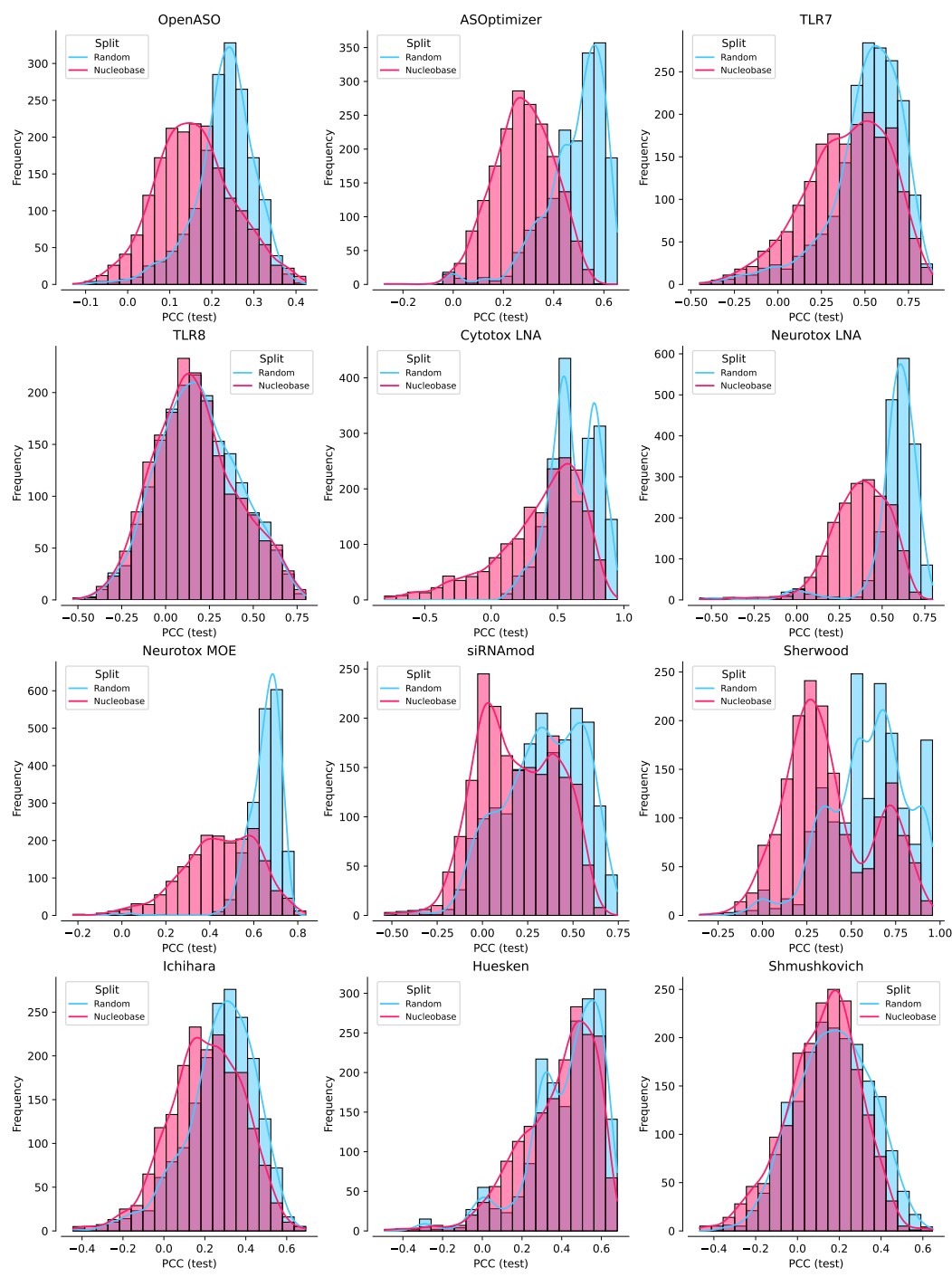

Figure A5: Distributions by dataset of all the PCC values obtained across configurations when using random cross-validation or nucleobase splits.

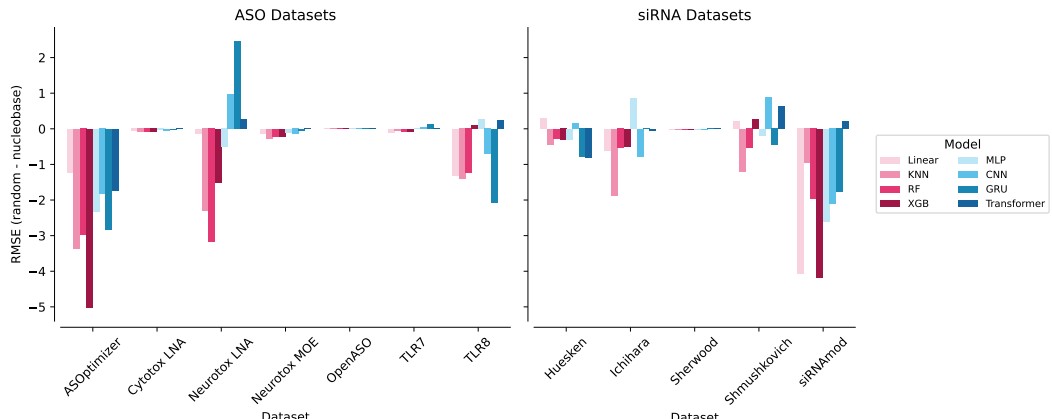

Figure A6: Differences in test RMSE evaluated with a random CV and nucleobase splits for the best performing models for each dataset.

## A.4 Graph Neural Network

Graph neural network (GNN) has recently become a popular deep learning architecture for small molecule modeling. For molecular property prediction, atoms are represented as nodes while bonds are represented as edges, and a GNN model uses a message-passing mechanism to aggregate feature information from neighbouring nodes. These updated node features are then aggregated into a single graph-level feature vector that represents the entire molecule. This graph-level representation is then used for the prediction of the target property. Here we adapt this approach for the modeling of oligonucleotide polymer and benchmark the performance of GNN for oligonucleotide property prediction.

### A.4.1 Featurizer

To prepare the oligonucleotide sequences for GNN model, we developed a custom featurizer that converts the HELM representation of each molecule into a graph structure. Unlike traditional atom-level molecular graphs, our approach employs a coarse-grained representation where each node corresponds to a nucleotide monomer component: a sugar, a base, or a phosphate group. This method models the oligonucleotide's chemical topology by creating distinct nodes for each component and connecting them based on their covalent bonds. Specifically, A sugar-phosphate backbone is formed by creating edges between the phosphate node of one monomer and the sugar node of the subsequent monomer. Each base node is connected as a branch to its corresponding sugar node. Each sugar node is also connected to its corresponding phosphate node within the same monomer unit. The features for each node is the one-hot encoded monomer identity. The final output for each oligonucleotide is a graph vector containing a node feature matrix and an edge index matrix. This coarse-grained featurization allows the model to learn hierarchical patterns based on the fundamental building blocks of the oligonucleotide polymer rather than individual atomic interactions.

### A.4.2 Model

We implemented a Graph Convolutional Network (GCN) architecture using the PyTorch Geometric python library. The model processes the input graph through a stack of GCN layers, where each layer updates a node's feature vector by aggregating information from its immediate neighbors, followed by a ReLU activation. After the final GCN layer, a global pooling operation aggregates all the node embeddings into a single, fixed-size vector that represents the entire oligonucleotide. This graph-level embedding is then fed into a fully connected linear layer to yield the final prediction for the target property.

### A.4.3 Experimental setup

We utilize the same experimental set up as for the models presented in the main text for benchmarking GNN model on the OligoGym dataset. Specifically we perform a grid-search of key hyperparameters

and evaluated each hyperparameter combination using a range of regression metrics on a standard 5-fold random cross validation and a 5-fold repeated nucleobase split. The hyperparameter grid used in this study is comprised of number of hidden dimension in each graph convolution layer (32, 64), number of graph convolution layers (1, 2) and the choice of global pooling operations (mean, max, sum). All models were trained with the Adam optimizer for maximum of 100 epochs. We used early stopping with patience of five epochs to avoid overfitting.

### A.4.4 Performance

We report the performance of the best GNN model on the random and nucleobase split in Table A13. The GNN models did not yield improved performance for any dataset in OligoGym compared to sequence-based deep learning models or classical machine learning models. The lack of improvement observed with GNN models is likely attributable to the sequential nature of oligonucleotide graphs. Unlike the more complex structural graphs of small molecules, oligonucleotide graphs are relatively simple linear polymer graphs. Consequently, a GNN architecture offers minimal advantages compared to sequence models for this type of data. A promising future study is to use a more fine-grained representation of oligonucleotides, for example at the atomic level using the SMILES representation provided in the datasets. This could enable a more detailed understanding of the chemical nature of these modified oligonucleotide and could lead to improved performance.

Table A13: Summary of GNN performance, PCC.

| Splitting strategy
Dataset | Random | Nucleobase |
|---|---|---|
| OpenASO | $0.27 \pm 0.02$ | $0.20 \pm 0.02$ |
| ASOptimizer | $0.37 \pm 0.01$ | $0.33 \pm 0.09$ |
| TLR7 | $0.52 \pm 0.10$ | $0.52 \pm 0.15$ |
| TLR8 | $0.32 \pm 0.15$ | $0.24 \pm 0.28$ |
| Cytotox LNA | $0.76 \pm 0.03$ | $0.65 \pm 0.16$ |
| Neurotox LNA | $0.55 \pm 0.04$ | $0.52 \pm 0.16$ |
| Neurotox MOE | $0.67 \pm 0.03$ | $0.60 \pm 0.09$ |
| siRNAmod | $0.47 \pm 0.07$ | $0.46 \pm 0.06$ |
| Sherwood | $0.27 \pm 0.00$ | $0.23 \pm 0.17$ |
| Ichihara | $0.31 \pm 0.09$ | $0.35 \pm 0.11$ |
| Huesken | $0.36 \pm 0.04$ | $0.34 \pm 0.06$ |
| Shmushkovich | $0.11 \pm 0.09$ | $0.12 \pm 0.10$ |

### A.5 RNA-FM embeddings

A common strategy to improve predictive performance in low-data regimes is transfer learning, where embeddings from large-scale foundation models are used as inputs to smaller architectures—a process often referred to as fine-tuning.

In this work, we evaluated whether such an approach could provide benefits by conducting a small-scale comparison. Specifically, we trained a convolutional neural network (CNN) on two types of input representations: (i) one-hot-encoded nucleotide sequences and (ii) nucleotide-level embeddings generated by the widely used RNA-FM foundation model [62]. RNA-FM is pretrained primarily on non-coding RNA sequences from the RNAcentral database, and its embeddings have been shown to enhance performance in other RNA benchmarks [19, 20]. For consistency, the CNN was fixed to a single set of hyperparameters (D=2, H=64, W=5, P=max), the experimental setup was otherwise the same as that used for the large benchmarking experiment. Results are summarized in Tab. A14.

Overall, the use of RNA-FM embeddings offered little to no improvement over the one-hot baseline. In fact, for certain datasets (e.g., Cytotox LNA and siRNAmod), the one-hot representation performed substantially better. This outcome is unsurprising: RNA-FM does not account for chemically modified nucleotides, so the input was limited to natural analogs, thereby missing all the information about chemical modifications that are known to modulate efficacy and toxicity. The observed patterns closely mirror those in Fig. 1, where we compare the performance of models when trained with a one-hot-encoding with and without chemical modification information.

While transfer learning from large foundation models remains a promising research direction, our findings highlight a key limitation for oligonucleotide modeling. Models that cannot represent chemical modifications may fail to provide a clear advantage, whereas models trained purely on chemical representations such as SMILES strings may lack the abstraction needed to capture the properties of long, linear nucleotide chains. Future advances in this space will likely require hybrid approaches that integrate the strengths of both sequence-based foundation models and chemically informed models to achieve more expressive and robust representations.

Table A14: Comparison of different featurizers on the ASO and siRNA datasets.

| Featurizer
Dataset | OneHotEncoder | RNA-FM Embeddings |
|---|---|---|
| OpenASO | 0.28 ± 0.04 | 0.28 ± 0.02 |
| ASOptimizer | 0.55 ± 0.01 | 0.54 ± 0.01 |
| TLR7 | 0.59 ± 0.17 | 0.59 ± 0.11 |
| TLR8 | -0.01 ± 0.24 | 0.02 ± 0.14 |
| Cytotox LNA | 0.88 ± 0.02 | 0.58 ± 0.06 |
| Neurotox LNA | 0.64 ± 0.04 | 0.66 ± 0.06 |
| Neurotox MOE | 0.73 ± 0.02 | 0.7 ± 0.03 |
| siRNAmod | 0.56 ± 0.05 | 0.21 ± 0.07 |
| Sherwood | 0.84 ± 0.0 | 0.82 ± 0.01 |
| Ichihara | 0.45 ± 0.08 | 0.48 ± 0.15 |
| Huesken | 0.63 ± 0.02 | 0.59 ± 0.04 |
| Shmushkovich | 0.24 ± 0.1 | 0.23 ± 0.14 |

Table A15: Summary of model performance evaluated with nucleobase splits. The results of the best performing configuration for each model class are shown. Each cell contains the mean and standard deviation of the PCC across splits.

| Model Dataset | Linear | KNN | RF | XGB | MLP | CNN | GRU | GNN | Transformer |
|---|---|---|---|---|---|---|---|---|---|
| OpenASO | **0.29 ± 0.08** | 0.23 ± 0.06 | 0.29 ± 0.03 | 0.21 ± 0.02 | 0.23 ± 0.09 | 0.25 ± 0.08 | 0.19 ± 0.05 | 0.20 ± 0.02 | 0.27 ± 0.11 |
| ASOptimizer | 0.39 ± 0.04 | 0.45 ± 0.06 | 0.48 ± 0.03 | 0.40 ± 0.06 | 0.43 ± 0.08 | **0.48 ± 0.04** | 0.41 ± 0.03 | 0.33 ± 0.09 | 0.44 ± 0.08 |
| TLR7 | 0.68 ± 0.07 | 0.56 ± 0.13 | **0.73 ± 0.10** | 0.66 ± 0.07 | 0.66 ± 0.11 | 0.62 ± 0.17 | 0.32 ± 0.13 | 0.52 ± 0.15 | 0.72 ± 0.11 |
| TLR8 | 0.59 ± 0.08 | 0.41 ± 0.13 | **0.70 ± 0.06** | 0.58 ± 0.16 | 0.57 ± 0.10 | 0.25 ± 0.19 | 0.13 ± 0.13 | 0.24 ± 0.28 | 0.26 ± 0.20 |
| Cytotox LNA | 0.64 ± 0.16 | 0.64 ± 0.07 | 0.69 ± 0.09 | 0.64 ± 0.15 | **0.72 ± 0.11** | 0.70 ± 0.11 | 0.32 ± 0.23 | 0.65 ± 0.16 | 0.72 ± 0.06 |
| Neurotox LNA | 0.56 ± 0.09 | 0.49 ± 0.09 | 0.56 ± 0.08 | 0.50 ± 0.16 | 0.53 ± 0.08 | **0.57 ± 0.08** | 0.17 ± 0.22 | 0.52 ± 0.16 | 0.53 ± 0.10 |
| Neurotox MOE | 0.59 ± 0.24 | 0.49 ± 0.05 | 0.61 ± 0.11 | 0.58 ± 0.04 | 0.66 ± 0.07 | 0.67 ± 0.10 | 0.67 ± 0.12 | 0.60 ± 0.09 | **0.70 ± 0.06** |
| siRNAmod | 0.44 ± 0.11 | 0.50 ± 0.09 | **0.53 ± 0.09** | 0.42 ± 0.14 | 0.47 ± 0.07 | 0.49 ± 0.12 | 0.25 ± 0.09 | 0.46 ± 0.06 | 0.49 ± 0.09 |
| Sherwood | 0.70 ± 0.05 | 0.73 ± 0.03 | 0.82 ± 0.03 | **0.90 ± 0.03** | 0.74 ± 0.04 | 0.74 ± 0.04 | 0.12 ± 0.10 | 0.23 ± 0.17 | 0.44 ± 0.09 |
| Ichihara | 0.49 ± 0.09 | 0.38 ± 0.14 | 0.49 ± 0.06 | 0.34 ± 0.13 | **0.57 ± 0.12** | 0.48 ± 0.10 | 0.21 ± 0.10 | 0.35 ± 0.11 | 0.35 ± 0.14 |
| Huesken | 0.62 ± 0.01 | 0.44 ± 0.01 | 0.59 ± 0.01 | 0.51 ± 0.03 | 0.62 ± 0.01 | **0.63 ± 0.03** | 0.16 ± 0.04 | 0.34 ± 0.06 | 0.51 ± 0.09 |
| Shmushkovich | 0.20 ± 0.08 | 0.28 ± 0.16 | **0.42 ± 0.14** | 0.35 ± 0.07 | 0.35 ± 0.04 | 0.27 ± 0.10 | 0.12 ± 0.11 | 0.12 ± 0.10 | 0.17 ± 0.20 |

