# OpenReview forum: "OligoGym: Curated Datasets and Benchmarks for Oligonucleotide Drug Discovery"
_NeurIPS.cc/2025/Datasets_and_Benchmarks_Track — NeurIPS 2025 Datasets and Benchmarks Track poster_

### Official Review · Reviewer_tiUe · 2025-06-29

**Rating:** 5
**Confidence:** 3

**Summary:**

This work introduces OligoGym, a novel collection of ML ready datasets for oligonucleotide therapeutics. The main contributions of the work includes the creation of the datasets which contains  diverse oligonucleotide therapeutic datasets across multiple modalities and endpoints to address the data scarcity and inconsistency in current literature. Another contribution is the benchmarking efforts which includes classical and deep learning baselines such as CNNs and MLPs. Lastly, the authors provided clear tutorial on how to download and process dataset from their github.

**Additional Feedback:**

- The paper and repo is submitted as single blind however the paper is formatted as double blind, please be consistent in the format that you choose.
- Discussing one hot encoding in Section 4.1.2 is unnecessary for the main paper and should go in the appendix.

**Dataset Code Accessibility:**

Yes

**Dataset Code Comments:**

Yes, I have checked the provided github and saw clear documentation as well as tutorials.

**Ethical Comments:**

No ethical concerns

**Ethical Considerations:**

No, there are no or only very minor ethics concerns

**Final Justification:**

The authors have adequately addressed my concerns. I believe the benchmark will serve as a useful tool for the research community. Therefore, I recommend accept.

**Limitations Weaknesses:**

Here are my concerns, mainly related to the baseline selection and current empirical results
- **lack of transformer baseline**: given the recent success of transformer architecture, it should be included as a baseline.
- **surprising finding**: It is surprising that Random Forest has the overall best empirical performance. More explanations are needed why do we expect random forest to perform best?
- **questions regarding hyperparameter tuning**: the authors mentioned tuning on the features however, it is not clear how much effort went into tuning the model architectures of CNN and MLP for this task, potentially with hyperparameter search, the performance of these two baselines might improve significantly.

**Strengths Contributions:**

Here are the strengths of this submission:
- **addressing gaps in literature**: the proposed benchmarks clearly addresses gaps in the current datasets including lack of toxicity end points and datasets for chemically modified oligos
- **standardized dataset and evaluation**: the authors proposed easy to use code to standardize evaluation for Oligonucleotide therapeutics, addressing inconsistencies in the literature. The authors also introduced novel human readable XNA data format.

- **clear writeup and documentation**: the paper is easy to follow and the code documentation is also clear and intuitive.

---

> ### Author Rebuttal · Authors · 2025-07-31
>
> We thank the reviewer for an insightful discussion of the manuscript and appreciate the positive response to the work. We include a point-by-point discussion of each question raised by the reviewer below.
>
> > Q1
>
> The reviewer raises a great point, it is true that the transformer architecture has enabled tremendous improvements for numerous machine learning tasks in the field. We report here the results including a Transformer architecture which takes as input the one-hot-encoded representation of the compounds in our datasets. In order to satisfy the request from another reviewer we also include results from a graph neural network. The hyperparameters of both architectures were tuned for each dataset with the same procedure used for the main benchmarking experiment in our manuscript. In the table below we report the mean test Pearson correlation evaluated using random split cross-validation.
>
> | Dataset         | Linear     | KNN        | RF           | XGB         | MLP        | CNN        | GRU        | GNN        | Transformer |
> |----------------|------------|------------|--------------|-------------|------------|------------|------------|------------|-------------|
> | OpenASO        | 0.33 ± 0.02 | 0.32 ± 0.01 | **0.35 ± 0.04** | 0.27 ± 0.02 | 0.27 ± 0.03 | 0.30 ± 0.03 | 0.18 ± 0.02 | 0.26 ± 0.03 | 0.30 ± 0.03 |
> | ASOptimizer    | 0.47 ± 0.01 | 0.63 ± 0.01 | **0.64 ± 0.01** | **0.64 ± 0.01** | 0.58 ± 0.01 | 0.56 ± 0.02 | 0.57 ± 0.01 | 0.38 ± 0.01 | 0.54 ± 0.01 |
> | TLR7           | 0.77 ± 0.02 | 0.60 ± 0.09 | **0.78 ± 0.04** | 0.74 ± 0.07 | 0.72 ± 0.08 | 0.66 ± 0.09 | 0.35 ± 0.22 | 0.56 ± 0.05 | 0.76 ± 0.04 |
> | TLR8           | 0.54 ± 0.13 | 0.42 ± 0.14 | **0.68 ± 0.08** | 0.59 ± 0.14 | 0.53 ± 0.12 | 0.19 ± 0.07 | 0.16 ± 0.16 | 0.27 ± 0.37 | 0.34 ± 0.31 |
> | Cytotox LNA    | 0.79 ± 0.02 | 0.88 ± 0.03 | **0.91 ± 0.02** | 0.91 ± 0.01 | 0.86 ± 0.01 | 0.88 ± 0.02 | 0.49 ± 0.05 | 0.73 ± 0.03 | 0.73 ± 0.04 |
> | Neurotox LNA   | 0.64 ± 0.03 | 0.68 ± 0.03 | **0.76 ± 0.02** | 0.71 ± 0.03 | 0.63 ± 0.03 | 0.67 ± 0.04 | 0.42 ± 0.21 | 0.54 ± 0.02 | 0.63 ± 0.04 |
> | Neurotox MOE   | 0.70 ± 0.02 | 0.69 ± 0.01 | 0.74 ± 0.02 | 0.73 ± 0.02 | 0.73 ± 0.04 | **0.75 ± 0.02** | 0.73 ± 0.02 | 0.66 ± 0.02 | **0.75 ± 0.02** |
> | siRNAmod       | 0.62 ± 0.06 | 0.56 ± 0.04 | **0.69 ± 0.03** | 0.68 ± 0.03 | 0.62 ± 0.07 | 0.62 ± 0.04 | 0.17 ± 0.11 | 0.46 ± 0.06 | 0.45 ± 0.06 |
> | Sherwood       | 0.78 ± 0.00 | 0.85 ± 0.00 | 0.92 ± 0.00 | **0.96 ± 0.00** | 0.82 ± 0.00 | 0.85 ± 0.00 | 0.13 ± 0.00 | 0.27 ± 0.00 | 0.59 ± 0.06 |
> | Ichihara       | 0.54 ± 0.07 | 0.38 ± 0.07 | 0.49 ± 0.02 | 0.38 ± 0.08 | 0.54 ± 0.07 | **0.56 ± 0.02** | 0.18 ± 0.14 | 0.30 ± 0.11 | 0.37 ± 0.05 |
> | Huesken        | 0.63 ± 0.02 | 0.45 ± 0.02 | 0.62 ± 0.04 | 0.53 ± 0.03 | 0.64 ± 0.01 | **0.65 ± 0.02** | 0.14 ± 0.03 | 0.36 ± 0.05 | 0.55 ± 0.02 |
> | Shmushkovich   | 0.24 ± 0.09 | 0.38 ± 0.07 | **0.50 ± 0.08** | 0.41 ± 0.07 | 0.30 ± 0.21 | 0.27 ± 0.08 | 0.05 ± 0.06 | 0.11 ± 0.08 | 0.17 ± 0.07 |
>
> The transformer model overall performed similarly to the CNN, in some cases outperforming it (e.g. TLR7). Nonetheless the CNN model remained the best performing deep learning model on our datasets. We think this can be attributed to the relatively short length of oligonucleotide compounds. The ability of transformers to integrate information at longer ranges is not as beneficial as for other tasks with longer biological sequences. We will include these additional results in the final version of the manuscript.
>
> > Q2
>
> Although this finding may seem surprising, it is important to note that random forest models continue to deliver state-of-the-art performance on low-N datasets and remain a preferred choice for many predictive tasks involving tabular data [1]. Tree-based methods have also been shown to outperform neural networks when dealing with heavy-tailed distributions, which are commonly observed in biological datasets [2]. Furthermore, the observation that models relying on tabular data representations (such as Linear, KNN, RF, XGB, MLP) perform as well as, or nearly as well as, sequence-based models (such as CNNs and GRUs) suggests that the association between compound features (e.g., nucleotides, chemical modifications) and measured endpoints may not strongly depend on their positional context within a sequence.
>
> In order to clarify this aspect we will add the following statement at line 252 in the manuscript:
>
> “[Even when not the absolute best, random forests consistently achieved competitive results], confirming previous findings that such models can deliver state-of-the-art performance on small datasets [1].”
>
> And at line 258:
>
> “Interestingly, models trained on tabular representations of the data (Linear, KNN, RF, XGB, MLP) either outperformed or were on par with sequence-based models (CNN, GRU), suggesting that the association between compound features (e.g., nucleotides, chemical modifications) and measured endpoints may not strongly depend on their positional context within a sequence.”
>
> [1] Grinsztajn, Léo, Edouard Oyallon, and Gaël Varoquaux. "Why do tree-based models still outperform deep learning on typical tabular data?." Advances in neural information processing systems 35 (2022): 507-520.
>
> [2] McElfresh, Duncan, et al. "When do neural nets outperform boosted trees on tabular data?." Advances in Neural Information Processing Systems 36 (2023): 76336-76369.
>
> > Q3
>
> We agree with the reviewer about the importance of hyperparameter tuning, especially for deep learning architectures. In our benchmark we not only tuned the parameters related to the featurization methods but also various model hyperparameters for each of the implemented models. The hyperparameters search grid is made available in **Table A3**. The results shown in **Table 2** and **Table 3** are obtained from the best performing model configurations across hyperparameters. While the hyperparameter grid might not be completely exhaustive, we believe that it strikes a reasonable balance between establishing baseline models and keeping the number of runs at a manageable level. Furthermore, since for many of the datasets we operate in a low-data regime, we expect that increasing the model capacity might lead to overfitting and thus poorer performance on the test set. We will also add a table containing the optimal hyperparameters for each dataset and model in the final version of the paper, as it was requested from another reviewer as well. We can't currently include it here due to the character limit.

---

> > ### Comment · Reviewer_tiUe · 2025-08-04
> > **Official Comment by Reviewer tiUe**
> >
> > The authors have addressed my concerns in detail. I understand that in low data settings, random forest models can be very effective. This benchmark might also potentially provide a good test case for pre-trained models to transfer to this low data setting and potentially achieving better results than random forest. Thank you for the rebuttal, I will keep my score

---

> > > ### Author Response · Authors · 2025-08-05
> > >
> > > We thank the reviewer's assessment of the manuscript and rebuttal. We also agree that pre-trained models will serve as a useful technique in the future to deal with the data scarcity suffered by this modality. We encourage future research to explore this topic using OligoGym to benchmark representation or transfer learning techniques that would be better suited to this modality.

---

### Official Review · Reviewer_bLBZ · 2025-06-30

**Rating:** 3
**Confidence:** 4

**Summary:**

This work perform a benchmark study on regression tasks for $12$ datasets related to oligonucleotide drug discovery --- the so-called ``OligoGym''. These 12 curated datasets include antisense oligonucleotides (ASOs), siRNAs, and shRNAs, which cover efficacy, toxicity, and immunomodulation endpoints.

It also benchmarks a broad suite of classical and deep-learning models (linear, KNN, RF, XGBoost, MLP, CNN, GRU) across multiple featurization strategies and data-splitting protocols, establishing performance baselines for oligonucleotide property prediction.

**Dataset Code Accessibility:**

Yes

**Dataset Code Comments:**

The codes on the data loading and model training are publicly available.

**Ethical Considerations:**

No, there are no or only very minor ethics concerns

**Final Justification:**

All concerns have been thoroughly addressed; the authors’ detailed rebuttal is commendable. I am happy to raise my score to 4 in response.

Awarding a 4, I note that the dataset—while well-curated and clean—is rather standard. As the authors acknowledge, substantial domain knowledge has already been injected at the featurization stage, so generic ML methods can be applied directly, leaving limited room for future research that would embed domain expertise more deeply into specialized models.

**Limitations Weaknesses:**

1.  Though I agree that the datasets are diversely collected, I feel that the manuscript lacks clear and unified guidelines for the selection of the $12$ datasets, which makes the benchmark doubtful. The authors appear to aggregate related datasets indiscriminately into the 'gym' without clear justification. A more rigorous discussion in Section 3.2 would strengthen the methodology.

2. The mathematical modeling of these datasets and the related tasks is unclear. All the regression methods evaluated here seem to be broad-spectrum for general tasks. No domain knowledges are used in the modeling. It can be improved by adding a more detailed introduction about the special characteristics of these related tasks at the beginning of Section 4 or Section 4.2.

3. The benchmark models seem outdated. The most recent model GRU in this work was proposed in 2014, which is eleven years old. Modern models, like the GNN or transformer-based models, which are popular in other therapeutic and drug discovery domains [1], but absent in this work. (I acknowledge that I am not an expert in this field, but a fast search on Google Scholar will reveal a lot of works about this issue)

[1] Gogoshin G, Rodin A S. Graph neural networks in cancer and oncology research: Emerging and future trends[J]. Cancers, 2023, 15(24): 5858.

**Strengths Contributions:**

1. Standardized evaluation efforts are made, including the sample-splitting, cross-validation and hyper-parameter selection.

2. Various models including parametric statistical models, tree-based models and modern NN-based models are evaluated.

3. The 12 datasets cover a diverse range of endpoints.

---

> ### Author Rebuttal · Authors · 2025-07-31
>
> We thank the reviewer for the thorough reviews of the work and appreciate the questions raised to strengthen the manuscript.  We include a point-by-point discussion of each question raised by the reviewer below.
>
> > Q1
>
> We understand the reviewer's concern about  the lack of justification for the selection of the datasets presented in OligoGym. We offer to include the following text to Section 3 to increase the clarity of the benchmark and the motivation behind it:
>
> “OligoGym's dataset collection is built upon the pre-clinical oligonucleotide drug discovery pipeline, which relies on various efficacy and safety benchmarks to advance molecules from initial screening to potential clinical trials. We identify on-target efficacy and toxicity profiles as crucial therapeutic endpoints influencing decisions across these stages. [1,2] On-target efficacy serves as the primary endpoint during hit identification to select compounds that effectively modulate their targets. Subsequently, in-vitro toxicity profiling is employed to de-risk candidate molecules and reduce the likelihood of clinical failures. We incorporate several toxicity endpoints representing common but distinct types of toxicity found in oligonucleotide programs. Cytotoxicity is a widely used general toxicity endpoint. [2] However, meeting this criterion alone does not guarantee success with other toxicity endpoints. Therefore, we include additional toxicity datasets such as immunomodulation to reflect the multifaceted safety hurdles that must be overcome in real-world oligonucleotide discovery programs.[2] Rather than concentrating on a single property type, OligoGym offers a diverse dataset collection designed to benchmark machine learning (ML) models capable of assessing the overall developability of oligonucleotide therapeutic candidates. Furthermore, we focused on collecting datasets with chemically-modified oligonucleotides as chemical modifications are major design elements for this modality and models that can handle these modifications are lacking in literature. OligoGym is intended to provide a training ground and assessment criteria for ML models that can significantly impact the multi-parameter optimization landscape of the oligonucleotide therapeutics discovery pipeline.”
>
> [1] Hagedorn, Peter H., et al. "Acute neurotoxicity of antisense oligonucleotides after intracerebroventricular injection into mouse brain can be predicted from sequence features." nucleic acid therapeutics 32.3 (2022): 151-162.
>
> [2] Goyenvalle, Aurélie, et al. "Considerations in the preclinical assessment of the safety of antisense oligonucleotides." Nucleic acid therapeutics 33.1 (2023): 1-16.
>
> > Q2
>
> The reviewer raises a good point about the need to incorporate domain knowledge into the modelling of these datasets. OligoGym represents the first effort to systematically and comprehensively benchmark machine learning models on oligonucleotide therapeutics endpoints.  This work aims to establish a simple but robust baselines using established machine learning architectures, hence the choice of methods that have been tried and tested on general tasks. Models like random forests and CNNs have also been widely used specifically for this task, many times without the implementation of specific featurization techniques for oligonucleotides [1, 2]. In our work, domain knowledge is introduced in the featurization methods, which are tailored to this drug modality and incorporate domain knowledge (e.g. inclusion of chemical modification information, k-mer featurization of RNA sequences). Furthermore, we also evaluate the models with nucleobase split in addition to the standard random cross-validation. This introduces a domain specific evaluation framework to ensure practical utility of the model in oligonucleotide drug discovery. We believe our benchmarks will prove valuable to further efforts in models that incorporate specific inductive biases for oligonucleotides to assess the improvement against these more basic models.
>
> To clarify this point we will add the following statement in section 4.2 at line 173:
>
> “[...] These models were chosen with the goal of establishing simple yet robust baselines, leveraging well-established architectures. Several of them have also been widely applied to specific tasks within this benchmark [1, 2]. Our evaluation framework further incorporates domain knowledge through the tailored featurization techniques described in the previous sections.”
>
> And at line 311 in our concluding remarks:
>
> “[...] The selection of machine learning models, while covering a broad range of approaches, was limited to architectures suited to general regression tasks. The benchmarking of models incorporating domain knowledge and specific inductive biases is an interesting avenue for future research.”
>
> [1] Hagedorn, Peter H., et al. "Acute neurotoxicity of antisense oligonucleotides after intracerebroventricular injection into mouse brain can be predicted from sequence features." nucleic acid therapeutics 32.3 (2022): 151-162.
>
> [2] Martinelli, Dominic D. "Machine learning for siRNA efficiency prediction: a systematic review." Health Sciences Review 11 (2024): 100157.
>
> > Q3
>
> We appreciate the reviewer concern regarding the lack of more contemporary architectures in the current benchmarks. We ran additional experiments with a transformer model with positional encoding as implemented in PyTorch and a graph convolutional neural network as implemented in PyTorch Geometric on all datasets. The transformer model was trained on the one-hot encoding of the compounds while the GNN was trained with a featurizer that converted HELM strings into RNA polymer graphs. In these graphs, each node represented a monomer identity, with phosphate and ribose nodes sequentially connected by edges to form the backbone. The nucleobase node branched off from each ribose at its corresponding position in the sequence. We present the results of this experiment in the table below, where we report the mean test Pearson correlation evaluated with a random split cross-validation.
>
> | Dataset         | Linear     | KNN        | RF           | XGB         | MLP        | CNN        | GRU        | GNN        | Transformer |
> |----------------|------------|------------|--------------|-------------|------------|------------|------------|------------|-------------|
> | OpenASO        | 0.33 ± 0.02 | 0.32 ± 0.01 | **0.35 ± 0.04** | 0.27 ± 0.02 | 0.27 ± 0.03 | 0.30 ± 0.03 | 0.18 ± 0.02 | 0.26 ± 0.03 | 0.30 ± 0.03 |
> | ASOptimizer    | 0.47 ± 0.01 | 0.63 ± 0.01 | **0.64 ± 0.01** | **0.64 ± 0.01** | 0.58 ± 0.01 | 0.56 ± 0.02 | 0.57 ± 0.01 | 0.38 ± 0.01 | 0.54 ± 0.01 |
> | TLR7           | 0.77 ± 0.02 | 0.60 ± 0.09 | **0.78 ± 0.04** | 0.74 ± 0.07 | 0.72 ± 0.08 | 0.66 ± 0.09 | 0.35 ± 0.22 | 0.56 ± 0.05 | 0.76 ± 0.04 |
> | TLR8           | 0.54 ± 0.13 | 0.42 ± 0.14 | **0.68 ± 0.08** | 0.59 ± 0.14 | 0.53 ± 0.12 | 0.19 ± 0.07 | 0.16 ± 0.16 | 0.27 ± 0.37 | 0.34 ± 0.31 |
> | Cytotox LNA    | 0.79 ± 0.02 | 0.88 ± 0.03 | **0.91 ± 0.02** | 0.91 ± 0.01 | 0.86 ± 0.01 | 0.88 ± 0.02 | 0.49 ± 0.05 | 0.73 ± 0.03 | 0.73 ± 0.04 |
> | Neurotox LNA   | 0.64 ± 0.03 | 0.68 ± 0.03 | **0.76 ± 0.02** | 0.71 ± 0.03 | 0.63 ± 0.03 | 0.67 ± 0.04 | 0.42 ± 0.21 | 0.54 ± 0.02 | 0.63 ± 0.04 |
> | Neurotox MOE   | 0.70 ± 0.02 | 0.69 ± 0.01 | 0.74 ± 0.02 | 0.73 ± 0.02 | 0.73 ± 0.04 | **0.75 ± 0.02** | 0.73 ± 0.02 | 0.66 ± 0.02 | **0.75 ± 0.02** |
> | siRNAmod       | 0.62 ± 0.06 | 0.56 ± 0.04 | **0.69 ± 0.03** | 0.68 ± 0.03 | 0.62 ± 0.07 | 0.62 ± 0.04 | 0.17 ± 0.11 | 0.46 ± 0.06 | 0.45 ± 0.06 |
> | Sherwood       | 0.78 ± 0.00 | 0.85 ± 0.00 | 0.92 ± 0.00 | **0.96 ± 0.00** | 0.82 ± 0.00 | 0.85 ± 0.00 | 0.13 ± 0.00 | 0.27 ± 0.00 | 0.59 ± 0.06 |
> | Ichihara       | 0.54 ± 0.07 | 0.38 ± 0.07 | 0.49 ± 0.02 | 0.38 ± 0.08 | 0.54 ± 0.07 | **0.56 ± 0.02** | 0.18 ± 0.14 | 0.30 ± 0.11 | 0.37 ± 0.05 |
> | Huesken        | 0.63 ± 0.02 | 0.45 ± 0.02 | 0.62 ± 0.04 | 0.53 ± 0.03 | 0.64 ± 0.01 | **0.65 ± 0.02** | 0.14 ± 0.03 | 0.36 ± 0.05 | 0.55 ± 0.02 |
> | Shmushkovich   | 0.24 ± 0.09 | 0.38 ± 0.07 | **0.50 ± 0.08** | 0.41 ± 0.07 | 0.30 ± 0.21 | 0.27 ± 0.08 | 0.05 ± 0.06 | 0.11 ± 0.08 | 0.17 ± 0.07 |
>
> Neither the transformer nor the GNN model outperformed other deep learning and classical machine learning approaches. The lack of improvement observed with GNN models is likely attributable to the sequential nature of oligonucleotide graphs. Unlike the more complex structural graphs of small molecules or proteins, oligonucleotide graphs are relatively simple linear polymer graphs. Consequently, a GNN architecture offers minimal advantages compared to sequence models for this type of data. The lack of improvement observed with the transformer model is also likely attributed to the more simplistic nature of oligonucleotide sequences. The maximum sequence length in our dataset is shorter than 30 nucleotides long, therefore the increased capacity that a transformer has to model long-range dependencies that have benefited other biological sequences like protein might not confer the same benefits for oligonucleotides. Furthermore, most of our datasets have relatively small sample sizes which might lead to overfitting when using a more complex architecture like the transformer. We will include these additional results and discussion in the appendix to further strengthen the benchmark as suggested. We will also update the github codebase with the GNN and transformer models, as well as the new graph-based featurizer to enable easy utilization of these models for oligonucleotide in the future.
>
> > Final remark
>
> We hope that we have adequately addressed all reviewer concerns and are open to further questions and feedback. We hope that our manuscript has been sufficiently improved to merit a reevaluation of its score.

---

> > ### Comment · Reviewer_bLBZ · 2025-08-03
> >
> > All concerns have been thoroughly addressed; the authors’ detailed rebuttal is commendable. I am happy to raise my score to 4 in response.
> >
> > Awarding a 4, I note that the dataset—while well-curated and clean—is rather standard. As the authors acknowledge, substantial domain knowledge has already been injected at the featurization stage, so generic ML methods can be applied directly, leaving limited room for future research that would embed domain expertise more deeply into specialized models.

---

> > > ### Author Response · Authors · 2025-08-05
> > >
> > > We thank the reviewer for the thorough assessment of the rebuttal and we appreciate the updated score. We understand the reviewer's concern but we believe the featurization techniques introduced in this study serve only as a starting point for further incorporation of domain knowledge in the modelling of this dataset such as the inclusion of target information, fine-grained chemical representation or structural information.

---

### Official Review · Reviewer_RMsm · 2025-07-01

**Rating:** 3
**Confidence:** 3

**Summary:**

This paper introduces OligoGym, a collection of standardized datasets for oligonucleotide drug discovery that encompasses various oligonucleotide therapeutic modalities and endpoints, and is ready for deploying machine learning models. The authors have used OligoGym to benchmark diverse classical and deep learning methods and featurization techniques.

**Additional Feedback:**

Please refer to the section on Limitations Weaknesses.

**Dataset Code Accessibility:**

Yes

**Dataset Code Comments:**

The authors have provided complete code and datasets in their GitHub repo: https://github.com/Roche/OligoGym.

**Ethical Considerations:**

No, there are no or only very minor ethics concerns

**Final Justification:**

During the rebuttal and discussion phase, I have engaged in some discussions with the authors on my previous concerns. Overall I think this work may still be of limited interest to researchers not on the specific field of oligonucleotide drug discovery. Despite including a novel and comprehensive dataset (for which I indeed acknowledge the authors' efforts), this work does not provide more insights on the model and hyper-parameter choices, so I still partially lean reject for this submission while I think it would be more suitable for other venues in bioinformatics.

**Limitations Weaknesses:**

- Since the application considered in this work is quite specific, the authors may need to convince researchers from other fields that their work can also be beneficial for them. Nevertheless, such discussion is missing in the current version.
- The baseline methods considered in this work seem not sufficient. Specifically, I wonder if RNA language models [1,2] can be applied to this task, as many oligonucleotide drugs share the same format as RNA sequences. Some discussions (preferably with some empirical results) should be welcome here.
- Also, there lack discussions on the impact of modality for different data sets. I wonder if two modalities (ASO and siRNA) introduced in this work can be transformed into each other, and if so, how will such modality affect the performance of different methods?
- Moreover, the exact choice of hyper-parameters are missing for all methods and data sets. It remains unclear to me if such hyper-parameters need to be specifically tuned for different data sets, which can limit the practical value of machine learning on this application.
- Finally, after reading the whole paper, I still cannot find any definitive conclusions on these machine learning models for oligonucleotide drug discovery. Specifically, I am still confused on which model and featurization approach generally perform the best, and it remains unclear to me how to interpret the model performance in Table 2 and 3.

## References

[1] Multi-purpose RNA language modelling with motif-aware pretraining and type-guided fine-tuning. Nature Machine Intelligence

[2] RNA language models predict mutations that improve RNA function. Nature Communication

**Strengths Contributions:**

The application (oligonucleotide drug discovery) looks novel and interesting.

---

> ### Author Rebuttal · Authors · 2025-07-31
>
> We thank the reviewer for the thorough reviews of the work and appreciate the questions raised to strengthen the manuscript.
>
> > Q1
>
> The reviewer made an important point about making the manuscript more interesting to the broader ML community. We will include an additional subsection to the conclusion with the following discussion:
>
> “OligoGym lowers the barrier to entry for machine learning researchers in oligonucleotide drug discovery by providing accessible, ML-ready datasets, introducing novel and unique challenges relevant to the broader ML community. Unlike natural biological sequences, oligonucleotide drugs rely on extensive chemical modifications to enhance their therapeutic profiles. This creates a complex interplay between biological sequence context and medicinal chemistry to achieve superior drug candidates. This complexity presents a unique challenge for the growing fields of multi-scale modeling and representation learning, a challenge also relevant to other therapeutic modalities, material science and climate modelling [1-3]. We ease future research in this direction by including both natural base and SMILES sequences in addition to the HELM sequences with modification information for all datasets. Furthermore, the synthetic nature of chemically modified oligonucleotides and the scarcity of datasets prevent the effective use of common transfer learning techniques employed in other biological domains, such as the use of foundation models pretrained on natural sequences. This necessitates further research into low-data and few-shot learning, an area that has also gained traction in other fields of drug discovery [4]. Finally, the multi-faceted nature of oligonucleotide drug discovery and the complex relationship between different properties (e.g., ASO efficacy and cytotoxicity) offer an interesting new challenge for multi-objective optimization research.”
>
> [1] Zhang, Zuobai, et al. "Multi-scale representation learning for protein fitness prediction." Advances in Neural Information Processing Systems 37 (2024): 101456-101473.
>
> [2] Kovachki, Nikola, et al. "Multiscale modeling of materials: Computing, data science, uncertainty and goal-oriented optimization." Mechanics of Materials 165 (2022): 104156.
>
> [3] Bodnar, Cristian, et al. "A foundation model for the Earth system." Nature (2025): 1-8.
>
> [4] van Tilborg, Derek, et al. "Deep learning for low-data drug discovery: Hurdles and opportunities." Current Opinion in Structural Biology 86 (2024): 102818.
>
> > Q2
>
> The reviewer raises an important point regarding the potential applicability of RNA language models to the tasks presented in our work. To explore this, we compared model performance using two different input representations: a one-hot encoding of the sequences and embeddings extracted from RNA-FM [1]. This approach is analogous to fine-tuning a frozen RNA-FM model on our specific tasks. We chose RNA-FM as it has achieved consistently high performance in other RNA benchmarks including RNAGym [2] and BEACON [3].
>
> A major limitation of using models trained solely on unmodified RNA sequences is the inability to incorporate information about chemical modifications. Consequently, we extracted nucleotide (token)-level embeddings only from the natural base sequences of the compounds in our datasets and used these as inputs to a CNN model (with fixed hyperparameters for simplicity). We chose a CNN architecture because it was the top-performing sequence model in our benchmarking study. The results of this comparison are shown in the table below, where we report the Pearson correlation on the training set after cross-validation on random splits.
>
> | Dataset         | CNN - OneHotEncoder | CNN - RNAFMEmbeddings |
> |-----------------|---------------------|------------------------|
> | OpenASO         | 0.28 ± 0.04         | 0.28 ± 0.02            |
> | ASOptimizer     | 0.55 ± 0.01         | 0.54 ± 0.01            |
> | TLR7            | 0.59 ± 0.17         | 0.59 ± 0.11            |
> | TLR8            | -0.01 ± 0.24        | 0.02 ± 0.14            |
> | Cytotox LNA     | 0.88 ± 0.02         | 0.58 ± 0.06            |
> | Neurotox LNA    | 0.64 ± 0.04         | 0.66 ± 0.06            |
> | Neurotox MOE    | 0.73 ± 0.02         | 0.70 ± 0.03            |
> | siRNAmod        | 0.56 ± 0.05         | 0.21 ± 0.07            |
> | Sherwood        | 0.84 ± 0.00         | 0.82 ± 0.01            |
> | Ichihara        | 0.45 ± 0.08         | 0.48 ± 0.15            |
> | Huesken         | 0.63 ± 0.02         | 0.59 ± 0.04            |
> | Shmushkovich    | 0.24 ± 0.10         | 0.23 ± 0.14            |
>
> Interestingly, we observe that both representations yield comparable performance across datasets. In some cases, one-hot encoding performs better (e.g., Cytotox LNA, siRNAmod), while in others, RNA-FM embeddings provide a stronger data representation (e.g., Ichihara). These trends closely mirror those shown in **Figure 1** of our manuscript, where we evaluate featurization techniques with and without chemical information.
> We believe that embeddings from large language models trained on RNA or DNA present a highly promising avenue for future research. However, it is clear that the value of such models will depend on their ability to integrate the chemical modification information that is critical to oligonucleotide drug discovery.
>
> [1] Chen, Jiayang, et al. "Interpretable RNA foundation model from unannotated data for highly accurate RNA structure and function predictions." arXiv preprint arXiv:2204.00300 (2022).
>
> [2] Arora, Rohit, et al. "RNAGym: Benchmarks for RNA Fitness and Structure Prediction." ICLR 2025 Workshop on Generative and Experimental Perspectives for Biomolecular Design.
>
> [3] Ren, Yuchen, et al. "Beacon: Benchmark for comprehensive rna tasks and language models." Advances in Neural Information Processing Systems 37 (2024): 92891-92921.
>
> > Q3
>
> We appreciate the reviewer’s comment and we realize the manuscript might benefit from a clearer description of the distinction between the two modalities. To this end, we will add this explanatory statement:
>
> Although both oligonucleotides, antisense oligonucleotides (ASOs) and short interfering RNAs (siRNAs) are fundamentally different therapeutic modalities that are not biochemically interchangeable due to their distinct modes of action, structure, and chemical modifications. ASOs are single-stranded molecules that typically recruit RNase H or cause steric hindrance. Their design often incorporates a "gapmer" architecture, as exemplified in the ASOptimizer, Cytotox LNA, and TLR7/8 datasets, featuring chemically modified flank regions (e.g., LNA or MOE) to increase stability and binding affinity, flanking a central DNA "gap" to trigger RNase H-mediated degradation of the target mRNA. In contrast, siRNAs (and shRNAs) are double-stranded molecules, composed of a guide and passenger strand, that are processed by the RISC pathway for mRNA target cleavage. Their modifications, such as the 2'-fluoro and 2'-O-methyl sugars detailed in the Shmushkovich and siRNAmod datasets, are optimized for RISC loading, stability, and in some cases, unassisted cellular uptake, which presents a completely different chemical and structural optimization problem to ASO.
>
> Given these differences in the biological mechanisms, sequence context and chemical design, we believe the two modalities are not directly transformable. Therefore, OligoGym provides the necessary datasets and codebase to facilitate the development of tailored predictive models for each specific modality.
>
> > Q4
>
> We agree with the reviewer that this is a gap in the manuscript. We want to clarify that we do have the information regarding the optimal hyperparameter choices, and a table listing the best hyperparameter configurations for each model and dataset is available in our public repository, within the notebook used to reproduce the results. This table shows that the optimal hyperparameter sets vary by dataset and was initially omitted due to formatting constraints with the publication, but we will include it in the final version of the manuscript. We can not attach it here due to the character limit.
>
> Regarding the need for dataset-specific tuning, the OligoGym benchmark includes datasets that vary in nature, feature set, label distribution, and size. Therefore, some degree of tuning is expected and often beneficial.
>
> > Q5
>
> We thank the reviewer for the feedback. We want to clarify that we are not attempting to claim that there is an overall better model or featurization applicable across all the presented datasets with this current manuscript. The goal of this work is to provide a benchmark resource that highlights performance variability and trends across a comprehensive set of model architectures, featurization techniques and data splits. The benchmarking experiment nonetheless presents some interesting trends that we highlight in the manuscript, for example:
>
> - Random forests consistently performed well across all datasets, and were the top performer model for 8 out of 12 datasets when evaluated with random splits
> - Deep learning approaches on the other hand outperformed classical ML methods for 6 out of 12 datasets when evaluated with nucleobase splits, showing higher potential for generalization to more challenging distribution shifts
>
> For what concerns featurization we found out that:
> - Excluding chemical modification information almost always results in poorer performance
> - Positional information is not of fundamental importance for the models in this benchmark
>
> We agree with the reviewer that model and featurizer performance is a critical question. Our work provides the first standardized platform to answer this question in a reproducible and extensible manner.
>
> > Final
>
> We hope that we have adequately addressed all reviewer concerns and are open to further questions and feedback. We hope that our manuscript has been sufficiently improved to merit a reevaluation of its score.

---

> > ### Comment · Reviewer_RMsm · 2025-08-06
> >
> > I would like to first thank the authors for their detailed responses. The authors are encouraged to incorporate all these discussions in their rebuttal into the revised version. While part of my previous concerns still remains (e.g., unclear conclusion on model and hyper-parameter choices), I will consider increasing my score to reflect the efforts in rebuttal.

---

> > > ### Author Response · Authors · 2025-08-06
> > >
> > > We thank the reviewer for the response to the rebuttal and appreciate the score increase. We will incorporate the changes discussed here in the final version of the manuscript.

---

### Official Review · Reviewer_eue3 · 2025-07-02

**Rating:** 5
**Confidence:** 3

**Summary:**

This paper presents OligoGym, the very first curated dataset for machine learning applications to oligonucleotide therapeutics discovery. The performance of various machine learning approaches (from classical to deep learning models) is benchmarked on this dataset. OligoGym makes the first step in creating a high-quality benchmark dataset for easing the research of machine learning approaches in oligonucleotide therapeutics applications.

**Additional Feedback:**

No additional feedback.

**Dataset Code Accessibility:**

Yes

**Dataset Code Comments:**

Accessible data and code (github) links are provided, sufficient dataset details are described in the paper, and code github readme page provides code running instructions.

**Ethical Comments:**

No ethical comments.

**Ethical Considerations:**

No, there are no or only very minor ethics concerns

**Final Justification:**

Considering the high quality and usefulness of the proposed OligoGym benchmark to biomedical domain application, and my concerns have been well addressed in rebuttal, I tend to believe this work is valuable to NeurIPS community and vote for acceptance.

**Limitations Weaknesses:**

As readers of NeurIPS papers are mostly not familar with the domain knowledge of biomedical science, authors are encouraged to add information about the following question:
- Are there any performance upper bounds for the oligonucleotide therapeutic tasks described in this paper?
- What performance do machine learning models need to reach so that they are practically useful in real-world oligonucleotide therapeutic applications?

**Strengths Contributions:**

- This paper makes a significant novelty contribution to fill a gap in the field of machine learning applications to oligonucleotide therapeutic drug discovery. The proposed high-quality dataset will be very useful to the broad researchers in this field.
- Comprehensive benchmarking of various machine learing models is conducted to ease the comparison to baseline performance.
- The writing of this paper is clear and well-organized.

---

> ### Author Rebuttal · Authors · 2025-07-31
>
> We thank the reviewer for an insightful discussion of the manuscript and appreciate the positive response to the work. We include a point-by-point discussion of each question raised by the reviewer below.
>
> > Q1
>
> The reviewer raises an excellent point with this question. Biological datasets are notoriously noisy and even the ideal model might not (and should not) be able to achieve perfect performance as measured by our metrics. The performance of any predictive model on such a noisy dataset has an upper bound defined by the signal-to-noise ratio in the dataset. Noise in biological datasets can be introduced by technical aspects related to the assay and protocol used to obtain the data or by biological aspects intrinsic to the biological pathways investigated, which might be stochastic by nature. One practical way to estimate the upper bound for the predictive performance of a machine learning model is to measure the correlation between technical or biological replicates.  In our datasets we unfortunately do not have a robust set of replicates for each compound to make such an estimate.
> In the Conclusion section we will insert the following statement at line 313:
>
> “Finally, the included datasets, since they were sourced from previous studies, might not have the required amount of technical and biological replicates needed to estimate an upper bound to the predictive performance that can be achieved by a predictive model. Biological datasets are notoriously noisy due to variability in assay protocols and the partially stochastic nature of biological processes, having multiple measurements for the same compound would enable better assessment of the amount and nature of the noise in a dataset.”
>
>
> > Q2
>
> The reviewer asks an important question regarding the necessary performance threshold for real-world applications. We will include the following discussion in the Conclusion section.
>
> “The performance threshold machine learning models must reach depends on the specific application and stage of discovery. During hit-identification an efficacy prediction model with a rank correlation of at least 0.7 could effectively prioritize the top 10-20% for experimental validation which typically screened around 1,000 to 2,000 compounds [1]. For lead optimization, where the goal is to precisely improve properties of a few lead candidates (~100s compounds), a performance metric such as RMSE is more appropriate to ensure low model error, with a target RMSE on par with the standard deviation of the experimental assay for comparable reliability [2]. For toxicity endpoints, where the goal is to minimize the risk of advancing toxic compounds, a recall metric of at least 0.95 (given a certain toxicity cutoff which depends on the toxicity type) would ensure this. The practical usefulness of an ML model must be assessed with a range of performance metrics appropriate for the discovery stage, many of which are provided in the OligoGym codebase.”
>
> Although we did not report the RMSE scores in the current manuscript, we do have access to the scores for all models and datasets, these are available in the github repository for future reference. We would also like to point out that we have already provided a range of selection metrics (e.g. top-k-recall) in the current codebase to help users assess the practical performance of their models in compound selection tasks.
>
> [1] Hagedorn, Peter H., et al. "Locked nucleic acid: modality, diversity, and drug discovery." Drug discovery today 23.1 (2018): 101-114.
>
> [2] Pradeep, Prachi, Katie Paul Friedman, and Richard Judson. "Structure-based QSAR models to predict repeat dose toxicity points of departure." Computational Toxicology 16 (2020): 100139.

---

> > ### Comment · Reviewer_eue3 · 2025-08-02
> > **Follow-up Response**
> >
> > I appreciate authors' efforts in rebuttal. All my concerns have been addressed so I will keep my acceptance decision.

---

> > > ### Author Response · Authors · 2025-08-05
> > >
> > > We thank the reviewer again for the reviews and assessment of the rebuttal. We are pleased to have addressed the reviewer's questions adequately and believe the manuscript will be stronger as a result.

---

### Note · Authors · 2025-08-12

We are grateful for the reviewers' insightful feedback on our manuscript and their evaluation of our rebuttals. We are confident that we have addressed all raised questions and, in doing so, have significantly strengthened the manuscript. We have incorporated additional discussion to clarify the purpose and impact of this benchmark. Following the reviewers' recommendations, we now include performance metrics for contemporary deep learning architectures, which, combined with our initial report on classical ML approaches, establish a robust performance baseline for future benchmarking. As many reviewers highlighted, OligoGym's primary contribution is the creation of a novel dataset collection, the first of its kind for oligonucleotide therapeutics, covering a wide array of endpoints. We hope this dataset, along with its accompanying codebase, will serve as a crucial first step in introducing this exciting new application domain to the machine learning community.

---

### Decision · Program_Chairs · 2025-09-18

**Decision:**

Accept (poster)

**Comment:**

OligoGym represents a curation of 12 datasets arising from the field of oligonucleotides, small synthetic RNA sequences, often designed with an eye toward therapeutic potential. To this end, datasets covering therapeutic efficacy, immunomodulation, and toxicity are presented here. The data is presented in HELM, SMILES, and base RNA sequences, with annotations for missing monomer information, and where possible, the target mRNA for each oligo. The work contributes a substantial amount of standardization across these datasets, a common barrier to entry for researchers interested in applying methods to biological domains such as these. The authors did a good job improving the accessibility of the text for a machine learning audience at the request of the reviewers. The reviewers reached broad consensus on this work: the work represents an interesting dataset in a niche domain, and that the dataset will be of use to researchers who want to work in this space, significantly lowering the barrier to entry. Simultaneously, the authors agree that the benchmarking element of this paper is not remarkable, and it is likely that more modern machine learning techniques working at the RNA sequence or chemical structure level will be able to improve on the results presented here out of the box. It is my judgement that the simplicity of the machine learning methods explored here is not very important - the value of the work lies in the dataset itself, and I am excited to see what work arises from the availability of this resource.

Note: reviewer RMsm and bLBZ mentioned they would update their score (both currently a 3), but failed to do so in the system. Reviewer bLBZ awarded a score of 4 (in text), RMsm was not clear. The other two reviews gave the paper a 5. I believe the authors addressed RMsm's concerns well. For this reason I am evaluating this paper as a 5/3/4/5 - 5/5/4/5 , since I cannot be sure of RMsm's final evaluation, which make this submission (in my estimation) closer to a 4.5 in aggregate score as a decent estimate of the true score (with 4.25 as a possible lower bound and 5 as an unlikely upper bound).